# Genetic Insights into Peripheral Artery Disease: A Narrative Review

**DOI:** 10.3390/biomedicines13112723

**Published:** 2025-11-06

**Authors:** Nonanzit Pérez-Hernández, José Manuel Rodríguez-Pérez, Luis Eduardo Nicanor-Juárez, Adriana Torres-Machorro, José Ramón García-Alva, Clara Villamil-Castañeda, Verónica Marusa Borgonio-Cuadra, Mirthala Flores-García

**Affiliations:** 1Department of Molecular Biology, Instituto Nacional de Cardiología Ignacio Chávez, Mexico City 14080, Mexico; unicanona@yahoo.com.mx (N.P.-H.); josemanuel_rodriguezperez@yahoo.com.mx (J.M.R.-P.); luisnicanorr1725@gmail.com (L.E.N.-J.); clara.vi.ca1@gmail.com (C.V.-C.); 2Escuela Superior de Medicina, Instituto Politécnico Nacional, Mexico City 11340, Mexico; 3Vascular Surgery Division, Department of Surgery, Instituto Nacional de Cardiología Ignacio Chávez, Mexico City 14080, Mexico; atorres.machorro@gmail.com (A.T.-M.); joseramon089@gmail.com (J.R.G.-A.); 4Programa de Doctorado en Ciencias Biomédicas, Universidad Nacional Autónoma de México, Mexico City 04510, Mexico; 5Laboratory of Genomic Medicine, Department of Genetics, Instituto Nacional de Rehabilitación Luis Guillermo Ibarra Ibarra, Mexico City 14389, Mexico; vborgoni@yahoo.com.mx

**Keywords:** peripheral artery disease, genetics, vascular disease, atherosclerosis, genetic susceptibility, genome-wide association studies

## Abstract

Peripheral arterial disease (PAD) is a complex, multifactorial atherosclerotic disease that primarily affects the arteries supplying the lower extremities, causing significant occlusion and reduced blood flow. Several studies have found an association between PAD and both genetic and environmental factors, which play a key role in the disease’s pathophysiology. Therefore, in this review, we describe the main genetic variants associated with plaque initiation, progression, and rupture in PAD. Furthermore, we identify different KEGG pathways involved in the pathological processes of these genes. We also describe gene expressions or transcriptomic studies, particularly in biopsies from patients with PAD. These findings could help identify the functional impact of genetic variants on the disease phenotype and, consequently, allow for the development of appropriate interventions that improve patient prognoses.

## 1. Introduction

Peripheral artery disease (PAD) is defined as an atherosclerotic arterial disease of the lower extremities [1,2]. It primarily affects arteries outside the aorta and the coronary arteries, predominantly those that supply the lower extremities, causing significant occlusion and reduction in blood flow [2].

The most commonly stenosed regions in the organism are vessels in the lower extremities, particularly the aortoiliac, femoropopliteal, and infrapopliteal vascular segments. PAD may manifest as intermittent claudication, defined as variable post-exertional pain that ceases with rest, or as chronic limb-threatening ischemia (CLTI) in advanced stages of the disease [3]. Notably, a significant number of individuals with PAD are either asymptomatic or have mild symptoms, often dismissing them as a normal part of aging. This perception can downplay the seriousness of PAD and contribute to its misrecognition [4,5,6].

In the evaluation of PAD, systematic lower-extremity examination is required, aiming to identify muscle or skin atrophy, loss of hair, and nail hypertrophy. Auscultation of bruits may be practical, but an exhaustive assessment of peripheral pulses is indispensable [4]. In appropriately selected patients, the ankle–brachial index (ABI) might be useful; this is a noninvasive test that calculates the ratio of systolic blood pressure (SBP) at the ankle to the SBP in the arm [4,6]. This test is used to assess vascular health, especially to detect PAD. A normal ABI value ranges from 1 to 1.4, borderline values are 0.91–0.99, and abnormal values are ≤0.9 for clinical and epidemiological purposes [4,6]. ABI is the standard method for diagnosing PAD; however, it is not reliable in patients with calcified noncompressible vessels, who tend to exhibit ABI values greater than 1.4.

This is commonly seen in individuals with diabetes and those with chronic kidney disease, who depend on the toe–brachial index (TBI) or more sophisticated imaging methods to establish an accurate diagnosis [4]. After diagnosis, additional imaging techniques (computed tomography angiography) may be necessary to properly assess the anatomical distribution of the disease and determine the most appropriate therapeutic approach [5,7].

For instance, individuals with vascular calcifications may exhibit an elevated ABI as a result of arterial stiffness, commonly referred to as medial calcification or Mönckeberg’s arteriosclerosis. This condition is prevalent among patients with diabetes mellitus or end-stage renal disease, rendering blood vessels less compressible and resulting in artificially elevated systolic pressures in the ankle compared to the arm, thus increasing the ABI above the normal range [8,9]. Recently, the Plantar Acceleration Time approach emerged as a reliable and novel technique that could serve as a valuable alternative in diabetic patients, especially when ABI and/or TBI are inconclusive or not applicable [10].

The recognition of PAD is of growing significance in light of the worldwide demographic shift toward an aging population. It is estimated that PAD affects over 230 million people globally. Studies indicate that PAD is slightly more common in younger individuals (40–60 years old) in low- and middle-income countries, but this prevalence rises sharply with age in high-income countries [6,11]. In this regard, by 2040, the elderly population is expected to increase by 22%, which will likely contribute to a significant healthcare burden due to atherosclerotic disease. Although evidence regarding sex differences in PAD is conflicting, men have traditionally been considered more prone to this disease; however, in recent years, there has been a notable rise in PAD cases among women, particularly in high-income countries.

The role of genetics for PAD is well known, as well as additional risk factors including, smoking, type 2 diabetes mellitus (DM2), hypertension, increased cholesterol levels, obesity, and older age [12].

Early efforts to study PAD genetics involved estimating heritability through family-based and twin studies, but these methods often overestimated the actual trait heritability. Subsequently, researchers turned to linkage analysis (LA), a traditional method centered on tracing disease-causing genes within families; however, this approach struggles to examine diseases such as PAD that involve polygenic interactions and environmental factors. As a result, LA has been replaced by association studies, such as candidate gene association studies (CGASs) and genome-wide association studies (GWASs) [7,13,14,15,16,17,18]. Therefore, the early detection of genetic factors associated with PAD is vital for the timely management of therapeutic approaches to reduce the growing healthcare burden. Thus, this narrative review presents a descriptive approach to identify the genetic determinants involved in the pathophysiology of PAD. We include the updated ESC 2024 PAD clinical guidelines to provide specific recommendations for appropriate disease management. We also identify the pathways involved and transcriptomics in genes that converge in PAD. Finally, we describe the relationship of genetic variants associated with PAD with environment, lifestyle, metabolic risk, and drugs.

## 2. Management of PAD in ESC 2024 Guidelines

The 2024 ESC Guidelines on the Management of Peripheral Arterial and Aortic Diseases (PAAD) underscore PAD as a chronic, progressive atherosclerotic condition associated with a significantly increased risk of cardiovascular events and limb-related complications. The guidelines emphasize the importance of early diagnosis, particularly through ABI screening in high-risk populations [Class I, Level B] as well as the comprehensive management of modifiable risk factors, including dyslipidemia, hypertension, DM2, and smoking [Class I, Level A]. Structured exercise therapy is also recommended as a core component of conservative treatment [Class I, Level A evidence]. Long-term follow-up within a multidisciplinary care framework is advised [Class I, Level C], and revascularization—either endovascular or surgical—should be considered in patients with critical limb-threatening ischemia (CLTI) [Class I, Level B] or non-healing ulcers. The need for revascularization and the risk of major amputation may now be stratified using validated clinical tools such as the WIfI (Wound, Ischemia, and foot Infection) classification system [Class IIa, Level C]. Future predictive models may now be stratified using validated clinical tools such as the WIfI classification system [Class IIa, Level C]. Future predictive models may incorporate genetic profiling to further refine risk stratification and personalize therapeutic decision-making in PAD management [2].

On the other hand, it is important to differentiate the genetic pattern for PAD rather than atherosclerosis. Recent studies emphasize the importance of differentiating PAD-specific genetic signatures from those driving coronary or cerebrovascular disease. Several methodological approaches may help delineate this specificity, including case-only analyses in PAD patients without CAD or stroke, conditional GWAS adjusting for shared atherosclerotic risk, and tissue-specific expression studies in peripheral arteries [19]. Transcriptomic and expression quantitative trait locus analyses suggest that certain variants influence gene expression in limb vasculature more strongly than in coronary or cerebral beds, pointing to localized mechanisms such as impaired angiogenesis and collateral formation [12]. Importantly, PAD progression is shaped by unique biomechanical and hemodynamic factors in peripheral circulation, such as shear stress, collateral vessel development, and ischemia–reperfusion dynamics, which may account for functional differences in the genetic determinants of disease expression. Thus, PAD genetics are characterized by both shared systemic pathways and modifiers with context-specific effects in the peripheral vasculature.

## 3. PAD Genetic Factors

PAD is not only shaped by lifestyle and aging, but also by genetic makeup. Some people are born with genetic variations that make their blood vessels more vulnerable to inflammation, cholesterol buildup, and clot formation. These inherited factors can silently increase the risk of developing PAD, especially when combined with conditions such as DM2 or high blood pressure. Understanding these genetic influences can help us detect the disease earlier and offer more personalized care to those at higher risk [20,21,22].

Recent genetic discoveries in PAD hold promise for translation into clinical practice. Several susceptibility loci, including *SH2B3* and *ABO*, have been linked to PAD and may eventually be integrated into polygenic risk scores alongside conventional risk factors, thereby improving patient stratification and early detection [12]. Beyond risk prediction, these findings open up the possibility of precision prevention, where individuals at elevated genetic risk could benefit from earlier vascular screening (e.g., ankle–brachial index, Doppler ultrasound) and the more aggressive management of modifiable risk factors [12]. Furthermore, the identification of molecular pathways unique to PAD may provide novel therapeutic targets, paving the way for personalized interventions that differ from strategies typically applied to coronary artery disease or stroke [23]. Figure 1 represents the main associated genes in PAD.

### 3.1. Variants Associated with Plaque Origination in PAD

The endothelium is a layer of cells that lines the interior of blood and lymphatic vessels and is essential for maintaining cardiovascular health and overall body function. It acts as a dynamic organ that regulates blood flow, inflammation, coagulation, and immune response [25]. A healthy endothelium produces substances that relax blood vessels (vasodilators). When endothelial dysfunction occurs, this ability is compromised, impairing blood flow. Endothelial dysfunction can trigger an inflammatory response in vessel walls, contributing to plaque formation and narrowing of the arteries [5,25].

Therefore, endothelial dysfunction and inflammation are central pathways to the pathogenesis of PAD. The endothelium plays a key role in regulating inflammation; however, in PAD, these processes become dysregulated, reducing nitric oxide (NO) production, increasing oxidative stress, and triggering harmful inflammatory responses. This contributes to vascular stiffness, plaque formation, and calcification. Inflammation then accelerates atherosclerosis progression by promoting arterial damage through cellular activation and the release of pro-inflammatory cytokines [5,25]. Genetic factors that regulate both endothelial function and inflammation are critical in understanding the molecular mechanisms leading to PAD. Variations in genes involved in NO synthesis, oxidative stress management, and inflammatory pathways offer the potential to identify novel markers and targets to restore vascular health and limit irreversible complications [26].

#### 3.1.1. NOS3

The endothelial nitric oxide synthase gene (*eNOS* or *NOS3*) is located on chromosome 7q36, which produces NO from L-arginine. NO plays a crucial role in vascular homeostasis by preventing atherogenesis and promoting vasodilation because it promotes the relaxation of blood vessels by reducing blood pressure. Moreover, it inhibits both platelet aggregation and adhesion. Genetic variants such as rs3918226 reduce eNOS activity, impairing NO production and disrupting vascular function. This polymorphism has been linked to vascular conditions such as PAD, where decreased NO bioavailability contributes to vascular dysfunction. Additionally, this variant has also been associated with low ABI values (OR = 2.86, 95% CI = 1.89–4.32, *p* < 0.0001), highlighting its role in PAD susceptibility [25,27,28].

#### 3.1.2. ICAM1

The *ICAM1* gene is located on chromosome 19p13.2 and encodes intercellular adhesion molecule 1 (ICAM-1), a glycoprotein with several functions that actively participates in cell adhesion, migration, and signaling, with a prominent role in the inflammatory response and the immune system. ICAM-1 may be a useful marker for detecting and evaluating inflammatory processes in the body as it mediates leukocyte adhesion during inflammation. The variant rs5498 has been associated with structural changes in ICAM-1, potentially increasing the risk of PAD by 67% among carriers [25,27,28].

#### 3.1.3. SELE

The *SELE* gene, located on chromosome 1q24.3, encodes E-selectin, a cell adhesion molecule expressed on cytokine-stimulated endothelial cells. E-selectin mediates the adhesion of leukocytes to the vascular lining, facilitating their migration to sites of inflammation, and plays a key role in atherosclerosis [29]. In addition, E-selectin’s involvement in leukocyte recruitment and vascular inflammation underscores its contribution to the development and progression of PAD. A genetic variant in *SELE*, rs5361, has been associated with an increased risk of PAD [30].

#### 3.1.4. LIPC

The *LIPC* gene, located on chromosome 15q21.3, encodes hepatic lipase (LIPC), an enzyme primarily expressed in the liver that plays a critical role in lipoprotein metabolism. LIPC hydrolyzes triglycerides and phospholipids in circulating lipoproteins such as high-density lipoprotein (HDL) and intermediate-density lipoprotein (IDL), facilitating their remodeling and clearance. This enzymatic activity is essential for maintaining cholesterol homeostasis and regulating plasma lipid levels. Genetic variants in *LIPC* can alter hepatic lipase activity, impacting the concentration and composition of atherogenic lipoproteins, further increasing the risk of vascular disease [31]. For instance, Ochoa et al. reported a specific variant (rs1800588) in the *LIPC* gene associated with elevated levels of these lipoproteins and an increased risk of PAD [30].

#### 3.1.5. LDLR

The *LDLR* gene, located on 19p13.2, encodes the low-density lipoprotein receptor (LDLR), which mediates the endocytosis of LDL and regulates cholesterol levels. This gene is linked to familial hypercholesterolemia, an autosomal-dominant disorder. The dysfunction or deficiency of LDLR causes LDL to accumulate in circulation, promoting atheroma formation [5,7,32,33]. Variants in the *LDLR* gene have also been associated with PAD [34].

#### 3.1.6. LPA

The *LPA* gene, located on chromosome 6q25.3-q26, encodes lipoprotein(a) [Lp(a)]. The pathogenicity of this protein is largely attributed to its Apo(a) subunit, where smaller isoforms promote oxidative reactions, increasing thrombotic risk and inhibiting plasmin activity [35,36]. Elevated Lp(a) serum levels are associated with an increased risk of atherosclerosis and vascular disease [37]. For instance, the variant rs118039278, associated with elevated Lp(a) concentrations, increases thrombotic risk and has been linked to PAD, highlighting its pathogenic role in vascular disease [7,38].

#### 3.1.7. LPL

The *LPL* gene, located on chromosome 8p21.3, encodes lipoprotein lipase (LPL), a key enzyme responsible for the hydrolysis of triglycerides in circulating lipoproteins such as chylomicrons and VLDL. LPL functions by releasing free fatty acids used for energy production or storage, and it also facilitates the uptake of lipoprotein remnants by interacting with endothelial cell surface proteins. Variants in the *LPL* gene can disrupt lipid metabolism, leading to elevated plasma triglycerides and contributing to vascular dysfunction. Notably, in GWAS related to PAD, rs322 in the *LPL* gene has been linked to altered lipid profiles and an increased risk of vascular disease, highlighting its importance in PAD pathophysiology [7,34,35].

#### 3.1.8. ABO

The *ABO* gene is located on 9q34.2 and encodes a glycosyltransferase that modifies the H antigen on the surface of red blood cells by adding specific sugar residues, thereby determining an individual’s ABO blood type (A, B, AB, or O). Beyond its hematologic role, the ABO glycosyltransferase also modifies glycoproteins and glycolipids expressed in endothelial cells, platelets, and other tissues, influencing intercellular interactions, inflammation, and thrombosis. It is suggested that variants in the *ABO* gene have been associated with a role in modulating lipid levels and influencing cardiovascular outcomes [39]. In this sense, *ABO* variant rs505922 has been tied to an increased PAD risk [7].

### 3.2. Variants Associated with Plaque Progression in PAD

Once plaques have formed, ongoing inflammation and changes in vascular structure promote their growth and complexity. Genetic factors that affect lipid metabolism contribute to fat accumulation inside the arterial walls, which worsens plaque features. At the same time, genes that regulate the structure and repair of the vessel walls may be altered, weakening the artery’s integrity. This can lead to the stiffening, calcification, and abnormal remodeling of the vessel, making plaques larger and more susceptible to rupture. Inflammation continues as immune cells release substances that break down the supporting tissue, further destabilizing plaques. The combined genetic influence on lipid metabolism, vessel structure, and inflammation drives this progression [19].

#### 3.2.1. IL-6

The *IL-6* gene, located on chromosome 7p15.3, encodes interleukin 6 (IL-6), a cytokine with important roles in inflammation, immune response, and B cell maturation. Moreover, IL-6 induces inflammatory responses through its receptor, IL6R, and is secreted at sites of acute and chronic inflammation. It plays a critical role in inflammatory diseases such as atherosclerosis, leading the production of other pro-inflammatory cytokines alongside IL-1 to amplify the immune response [40]. Furthermore, elevated levels of the IL-6 receptor, in either its membrane-bound form (IL6R) or soluble form (sIL6R), have been linked to a reduced risk of PAD, suggesting that genetic variants promoting these effects confer a protective profile [41].

#### 3.2.2. SH2B3

The *SH2B3* gene, located on chromosome 12q24.12, encodes the SH2B adaptor protein 3 (LNK), a key negative regulator of multiple signaling pathways involved in hematopoiesis, immune function, and vascular homeostasis by attenuating the JAK-STAT, MAPK, and PI3K pathways. When *SH2B3* is functioning correctly, it helps limit excessive immune responses and vascular cell proliferation, maintaining a balanced inflammatory environment. However, polymorphisms in *SH2B3* can impair this regulatory control, leading to dysregulated cytokine signaling and exaggerated inflammatory responses, promoting vascular injury, endothelial dysfunction, and smooth muscle proliferation; therefore, genetic variants have been linked to an increased susceptibility to PAD [42].

#### 3.2.3. CELSR2

The *CELSR2* gene encodes a member of the cadherin superfamily, which plays an important role in cell adhesion and receptor–ligand interactions for maintaining vascular tissue integrity. *CELSR2* is particularly important in regulating the structure and function of endothelial cells lining blood vessels. Genetic variations in *CELSR2* have been associated with altered levels of lipoprotein-associated phospholipase A2 (Lp-PLA2), an enzyme linked to vascular inflammation. Lp-PLA2 contributes to atherosclerosis by hydrolyzing oxidized phospholipids in LDL generating pro-inflammatory mediators that promote arterial wall inflammation and plaque instability. When *CELSR2* function is altered due to polymorphisms, Lp-PLA2 activity can be dysregulated, intensifying inflammatory processes in the arterial walls and accelerating endothelial dysfunction and plaque development [30,43].

#### 3.2.4. HDAC9

The *HDAC9* gene is located on chromosome 7p21.1 and encodes a protein that regulates transcription and cell cycle progression through histone deacetylation, influencing immune cell function. According to Klarin et al., genetic variants are linked to PAD, where inflammation contributes to vascular dysfunction and atherosclerosis. *HDAC9* may play a role in PAD by modulating inflammatory responses and smooth muscle cell proliferation, key factors in the disease’s progression [7,44].

#### 3.2.5. CDKN2B-AS1

The cyclin-dependent kinase inhibitor 2B antisense RNA 1 (CDKN2B-AS1) gene is located on chromosome 9p21 and encodes the long antisense noncoding RNA in the INK4 Locus (ANRIL) that regulates nearby genes involved in cell cycle control and vascular integrity, such as *CDKN2A* and *CDKN2B*. ANRIL interacts with chromatin-modifying complexes to silence these genes, helping maintain cellular balance. When ANRIL is dysregulated, it promotes vascular smooth muscle proliferation, impaired senescence, and chronic vascular inflammation—all key processes in plaque formation and disease progression. Studies have reported that rs1537372 is associated with an increased risk of atherosclerosis and PAD [7,38,45].

#### 3.2.6. PTPN11

This gene is localized on chromosome 12q24.13. *PTPN11* encodes a protein that is part of the protein tyrosine phosphatase (PTP) family. These proteins regulate several cellular processes, including cell growth, differentiation, and metabolism. It is expressed in many tissues and plays a role in cell signaling, cell migration, and transcription regulation. Mutations in this gene are linked to Noonan syndrome and acute myeloid leukemia. In addition, Klarin et al. and van Zuydam et al. found an association between the variant rs11066301 in this gene and PAD [7,38].

#### 3.2.7. CREB3L1

This gene is localized in humans on chromosome 11p11.2 and encodes a protein typically found in the endoplasmic reticulum (ER) membrane. When the ER experiences stress, this protein is cleaved, and its transcription factor domain moves to the nucleus where it activates the expression of target genes by binding to specific DNA elements. In the context of PAD, inflammation contributes to vascular injury, and the *CREB3L1* gene may indirectly influence these processes by regulating stress-induced inflammatory pathways. Recently, the genetic variant rs7476 in *CREB3L1* was associated with PAD [7,38].

### 3.3. Variants Associated with Plaque Rupture in PAD

In advanced stages, plaques can become unstable and rupture, exposing material that triggers clot formation. Genetic predisposition to increased blood clotting can raise the risk of thrombosis after rupture, leading to sudden blockages and ischemic events. Additionally, genetic factors affecting coagulation, platelet activity, and vessel wall health influence clot formation and stability. The interaction of chronic inflammation, endothelial dysfunction, and a prothrombotic tendency controlled by genetics ultimately shapes the clinical outcome in PAD patients [19].

Observational studies have reported an association between thrombotic events and worse clinical outcomes in patients with PAD. They often exhibit a prothrombotic state, which can exacerbate arterial blockages and increase the risk of complications such as limb ischemia and cardiovascular events. However, it remains unclear whether this relationship is directly causal or whether it might be influenced by reverse causation, where the severity of PAD itself leads to more thrombosis, or by confounding factors such as shared risk factors (e.g., DM2, smoking, inflammation). To clarify this, methods such as Mendelian randomization (MR) have been applied, aiming to assess causal links by using genetic variants as proxies for modifiable exposures. For instance, Klarin et al. used genetic data to investigate the causal role of coagulation factors in PAD risk but found that more research is needed to conclusively determine causality [19].

#### 3.3.1. COL4A1

The *COL4A1* gene is found on chromosome 13q34 and produces the alpha-1 chain of type IV collagen (COL4A1), an important building block of basement membranes. These membranes act like supportive scaffolds around blood vessels, helping to keep their structure strong and stable. The COL4A1 protein has special regions that allow it to form a network essential for maintaining the integrity of blood vessels. Beyond just providing support, this protein also helps remodel the walls of arteries and plays a role in controlling inflammation in the local environment [36]. Variants in the *COL4A1* gene can weaken these structures, leading to vascular problems such as PAD [7].

#### 3.3.2. SMOC1

The *SMOC1* gene is located on chromosome 14q24.2 and encodes SPARC-related modular calcium-binding protein 1, a multi-domain matricellular protein essential for ocular and limb development. Loss-of-function mutations in SMOC1 result in clinical conditions such as microphthalmia with limb anomalies [46]. Beyond developmental roles, GWASs have identified *SMOC1* as one of 19 genetic loci significantly associated with PAD, suggesting its potential contribution to vascular pathology. Although the specific molecular mechanisms remain to be elucidated, SMOC1’s known roles in extracellular matrix organization, the modulation of BMP/TGF-β signaling, and vascular calcification provide biological plausibility for its involvement in PAD pathogenesis [47].

#### 3.3.3. MMP3

The *MMP3* gene is located on chromosome 11q22.2-q22.3 and encodes matrix metalloproteinase-3 (MMP-3), a secreted enzyme that degrades key components of the extracellular matrix, including fibronectin, laminin, and collagen types III, IV, IX, and X. It also plays essential roles in tissue remodeling, wound repair, and the progression of atherosclerosis [48]. MMP-3 can also activate other MMPs, amplifying extracellular matrix breakdown and influencing vascular remodeling. Furthermore, elevated plasma levels of MMP-3 have been linked to an increased risk of PAD, supporting a causal role in disease progression through enhanced matrix degradation and vascular instability [44]. Genetic variants near or within *MMP3*, identified in large GWASs, show a statistically significant association with PAD (OR = 1.08; 95% CI: 1.05–1.11; *p* = 4.37 × 10^−9^) [19].

#### 3.3.4. F5

The *F5* gene, located on chromosome 1q24.2, encodes coagulation Factor V (FV), a critical protein in the blood coagulation cascade. When activated by thrombin, FV acts as a cofactor to accelerate the conversion of prothrombin to thrombin, which ultimately leads to fibrin clot formation. The rs6025 has been found to cause resistance to inactivation by activated protein C and is also strongly associated with an increased risk of venous thrombosis as well as arterial thrombosis, including PAD [7]. The rs6025 variant has also been linked to arterial thrombosis in multiple populations, underscoring its role in thrombophilia. Furthermore, recent studies suggest that additional variants in the *F5* gene may contribute to PAD complications such as chronic limb-threatening ischemia and the risk of amputation [30,49].

#### 3.3.5. F2

The *F2* gene, located on chromosome 11p11.2, encodes prothrombin, a precursor protein that is converted into thrombin, a serine protease critical for blood clot formation and maintaining vascular integrity. Thrombin plays multiple roles, including the conversion of fibrinogen to fibrin, the activation of platelets to promote aggregation, and involvement in tissue repair and angiogenesis. The rs1799963 variant in the *F2* gene has been linked to increased prothrombin levels and a heightened risk of thrombosis [50]. In PAD, carriers of the *A* allele show increased thrombin generation, which contributes to a prothrombotic state, raising the risk of PAD development and worsening vascular damage [49].

#### 3.3.6. FGB

The *FGB* gene is located on chromosome 4q31.3 and encodes the beta chain of fibrinogen, a glycoprotein essential for blood clot formation. Thrombin cleaves fibrinogen to form fibrin, which is crucial for clotting and vascular repair. Fibrinogen also regulates cell adhesion, vasoconstriction, and immune responses. Variants in *FGB* can lead to an increased thrombotic risk. In PAD, some variants in this gene raise plasma fibrinogen levels, which elevate the risk of PAD, stroke, and CAD. Fowkes et al., in their assessment of 121 PAD patients from the Edinburgh Artery Study, suggested that genetic variants of this gene are linked to an increased risk of PAD through structural changes in fibrinogen or interactions with nearby genes [51]. Table 1 shows the main genes and genetic variants involved in PAD.

To highlight the relevance of the genes described in Table 1 associated with PAD, a KEGG pathway enrichment analysis was performed to represent the pathways involved in these genes’ disease processes. In this regard, at the plaque formation stage, it allowed for the integration of the arginine biosynthesis pathway, the glycosphingolipid pathway, and the fluid shear stress and atherosclerosis pathway. Genes such as *NOS3*, *ABO*, and *SELE* are involved in these pathways (Figure 2a).

The importance of these pathways is as follows:(1).Arginine biosynthesis produces NO, a molecule that relaxes and dilates blood vessels, improving blood flow. In addition, L-arginine may improve walking ability and blood flow in these patients [56,57].(2).The biosynthesis of glycosphingolipids and their metabolites, when altered, can lead to their accumulation in tissues. This accumulation has been linked to the development and progression of PAD. It is suggested that alterations in sphingolipid metabolism contribute to the cellular and tissue damage that occurs during the atherosclerotic process [58].(3).Fluid shear stress and atherosclerosis, due to the force exerted by the constant flow of blood on the walls of blood vessels, play an important role in atherogenesis by altering the integrity of the endothelium, increasing its permeability, and allowing the entry of lipoproteins and inflammatory cells, which initiates the process of plaque formation and thrombi in the arteries and causes PAD [59].

Moreover, the pathways related to the stage of plaque progression include the PI3K-Akt signaling pathway, TNF signaling pathway, and insulin resistance pathway. Genes such as *NOS3*, *IL-6*, *CREB3L1*, *PTPN11*, and *SELE* are involved in these pathways (Figure 2b). The relevance of these pathways is as follows:(1).The PI3K-Akt signaling pathway is involved in regulating inflammation and oxidative stress, both of which are key factors in the pathogenesis of PAD [60].(2).The TNF signaling pathway promotes oxidative stress and decreases the bioavailability of NO, a crucial vasodilator, contributing to endothelial dysfunction. Furthermore, the TNF signaling pathway impacts vascular remodeling processes, leading to structural changes in the arteries of patients [61].(3).The insulin resistance pathway causes vascular damage through endothelial dysfunction via the inhibition of NO production. In addition, insulin resistance can activate pro-inflammatory molecular pathways, such as the MAP kinase (MAPK) pathway, which contribute to the disease [62].

Finally, the pathways related to the plaque rupture stage allow for the integration of complement activation, coagulation cascade, and platelet activation, which are involved in exacerbating the inflammatory process, which, together with endothelial dysfunction promotes thrombosis. Genes such as *F5*, *F2*, and *FGB* are involved in these pathways (Figure 2c) [63,64].

In parallel, we describe gene expression or transcriptomic studies that are crucial for the study of PAD. These studies are relevant as they demonstrate how changes in transcriptional activity could contribute to disease and reveal information not available at the genomic level. It is important to note that studies of this nature at the tissue level are scarce in PAD.

In this sense, transcriptomic and proteomic analyses performed in humans on gastrocnemius muscle biopsies from patients with PAD and control participants without PAD showed several enriched pathways, including those related to hypoxia, such as phosphatase and tensin homolog (PTEN), phosphoinositide 3-kinase (PI3K), and mitogen-activated protein kinase (MAPK) signaling [65].

Furthermore, WNT, Hedgehog, and Notch are among the key signaling pathways involved in the repair of damage caused by chronic hypoxia or ischemia/reperfusion. In this regard, a prolonged hypoxic environment in muscle is associated with induced mitochondrial damage, reduced ATP production, and the stimulation of inflammatory responses, including NF-κB activation [65]. It has also been shown that samples from patients with PAD are characterized by increased inflammation and decreased glycolysis compared to control participants without PAD [66].

On the other hand, transcriptomic studies in human gastrocnemius muscle biopsies showed a global overexpression of genes involved in stress response, autophagy, hypoxia, and muscle atrophy, as well as deficiencies in angiogenic protein signaling, response to misfolded proteins, and nerve repair, which could contribute to poor limb function in PAD [66,67]. Each of these mechanisms has potential for future research to develop effective interventions for the treatment of PAD.

## 4. Discussion

This review article involved recent contributions to PAD research from a genetics perspective.

All organisms are susceptible to variations in the DNA sequence, leading to genetic diversity. Genetic variants occur in different regions of genes, indicating that their product (a protein) will undergo structural or functional changes. Some diseases arise from mutations in a single gene, known as monogenic disorders, which follow Mendelian inheritance patterns. However, several diseases are polygenic, meaning that they are influenced by multiple genes, and they do not strictly adhere to inheritance. In fact, when both genetic and environmental factors contribute to the development of a disease, they fall under a different category, known as multifactorial or complex diseases [16,17]. For instance, PAD is clearly influenced by genetic variations and traditional metabolic risk factors. Table 2 presents genetic variants in different studies associated with PAD and their relationships with environmental and lifestyle factors, metabolic risks, and drugs.

Given the relevance of GWAS in PAD, Matsukura et al. identified three novel PAD susceptibility loci at *IPO5*/*RAP2A*, *EDNRA*, and *HDAC9* in a Japanese population. In particular, the *IPO5*/*RAP2A* locus revealed that rs9584669 conferred the risk of PAD. In addition, this variant showed reduced expression levels of IPO5 when validated through functional studies. This finding represented the first genetic risk evidence for PAD. Furthermore, few GWASs have identified the pathophysiological mechanisms associated with lipid metabolism, highlighting the genetic complexity of the disease [68].

In this regard, genetic association studies such CGASs and GWASs have identified variants associated with vascular diseases including PAD, small artery stroke, and atherosclerosis. Although this approach can reveal potential correlations, further analysis is necessary to validate their significance [14,15]. It is important to acknowledge that genetic heterogeneity is a well-recognized reason for the failure to replicate genetic association findings, including sample size, heterogeneity in the diagnostic criteria/definition of PAD, study power, study design, and the standardization of techniques and determinations. Table 3 depicts the potential causes of the differences observed in various studies.

Nevertheless, recent genetic discoveries in PAD hold promise for translation into clinical practice. Several susceptibility loci, including *SH2B3* and *ABO*, have been linked to PAD and may eventually be integrated into polygenic risk scores alongside conventional risk factors, thereby improving patient stratification and early detection [12]. Beyond risk prediction, these findings open up the possibility of precision prevention, where individuals at elevated genetic risk could benefit from earlier vascular screening (e.g., ankle–brachial index, Doppler ultrasound) and the better management of modifiable risk factors [12]. Furthermore, the identification of molecular pathways unique to PAD may provide novel therapeutic targets, paving the way for personalized interventions that differ from strategies typically applied to coronary artery disease or stroke [23].

## 5. Perspectives

The integration of genetic insights into PAD management has important clinical implications, particularly for risk stratification and personalized prevention. A proposed clinical pathway begins with the identification of at-risk individuals, combining family history and conventional factors with polygenic risk scores when available [104]. In those at elevated genetic risk, early non-invasive vascular screening (ankle–brachial index, duplex ultrasound) may be warranted, followed by intensified preventive strategies, including optimized lipid, blood pressure, and glycemic control, as well as structured smoking cessation. Genetic insights may also guide patient selection for novel therapies or clinical trials targeting PAD-specific molecular mechanisms. This stepwise integration of genetic risk into vascular practice exemplifies the potential of precision medicine to improve outcomes in PAD [12] (Figure 3).

Cardiovascular risk modeling focused on PAD prevention as well as pharmacogenomics applied to this context [106]. Monogenic risk models allow for early interventions in patients with genetic variants to prevent adverse outcomes, personalize therapy for comorbidities, and tailor the pathology to the specific mutation [107,108].

Furthermore, additional experimental strategies may be integrated in the research of gene functions by activating or silencing them. Therefore, gene editing in PAD is emerging as a promising but still developing and underexplored new field of research. Gene editing is being investigated through a gene therapy and genetic modification approach. This has made it an effective project for precisely modifying DNA. However, it is still an experimental therapy under development that could allow for the correction of genetic mutations or the silencing of harmful genes in the future, thus restoring vascular function in this group of patients. Although, this technology fits more for functional assay profiles; for example, recent studies have employed induced pluripotent stem cells as a model, utilizing CRISPR-Cas9 technology to introduce genetic variants into cell clones. This strategy might enable the functional validation of specific variants and the elucidation of their role as mechanisms in various complex diseases such as PAD [109,110,111,112].

## 6. Conclusions

Given its complexity, the future of PAD management lies partly in translational research. The study of genetic variants is crucial in this field, as they serve as a bridge between genetic knowledge and clinical applications. These genetic variants influence susceptibility to disease, response to treatment, and drug efficacy. They also act as markers that allow us to identify individuals at higher risk of developing complex diseases. By knowing an individual’s genetic risk, it is possible to implement prevention and early detection strategies to mitigate the impact of the disease. Despite advances in research, translation into clinical practice still presents challenges in terms of validation and accessibility. Overall, genetic analysis offers transformative opportunities for understanding and managing PAD. Continued efforts to conduct diverse large-scale studies and validate genetic tools in clinical settings will be crucial to fully realize the potential of genetic insights in PAD care in the era of molecular cardiology.

## Figures and Tables

**Figure 1 biomedicines-13-02723-f001:**
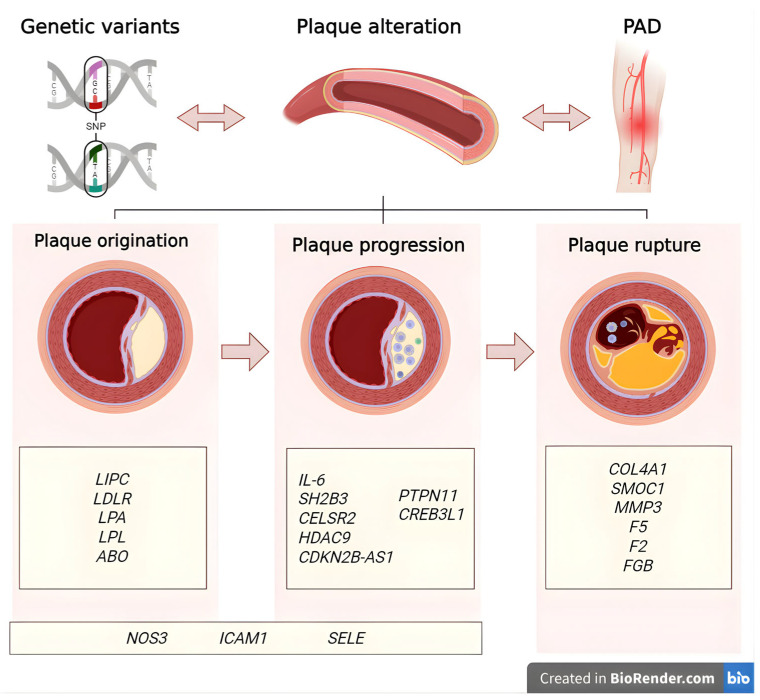
Gene classification by role in peripheral arterial disease. Genes are categorized by their predominant roles in the pathophysiology of peripheral arterial disease: those involved in plaque origination, progression through inflammatory and vascular remodeling processes, and plaque rupture leading to thrombotic events [24]. Created in BioRender. Villamil C (2025) https://app.biorender.com/illustrations/6368c2ffb888492c0c82d0c8 (accessed on 30 May 2025).

**Figure 2 biomedicines-13-02723-f002:**
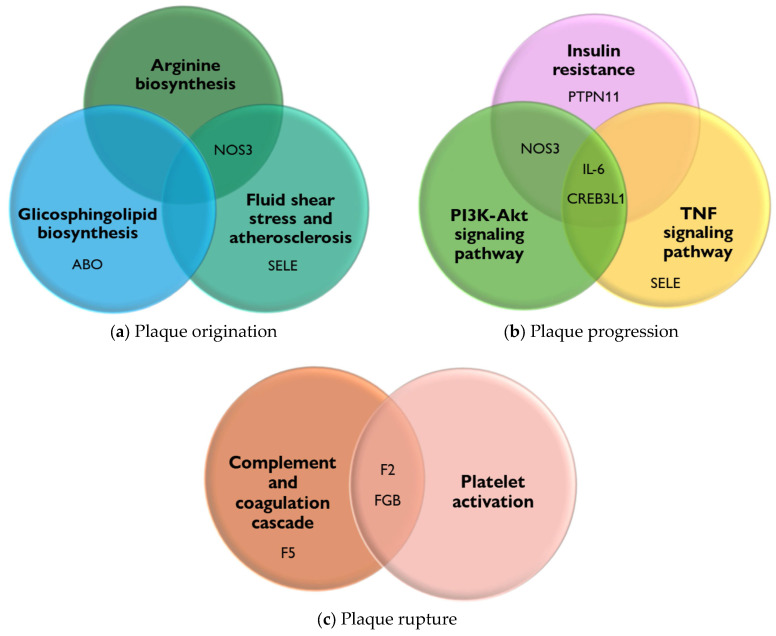
PAD genes and their connection to KEGG pathways. The PAD-associated genes involved in the origin, progression, and rupture of plaque were scrutinized using ExpressAnalyst (a web-based platform) for functional enrichment analysis to KEGG pathways. The overlap of genes involved in each pathway is highlighted [55]. (**a**) KEGG pathways interaction of plaque origination. (**b**) KEGG pathways interaction of plaque progression. (**c**) KEGG pathways interaction of plaque rupture.

**Figure 3 biomedicines-13-02723-f003:**
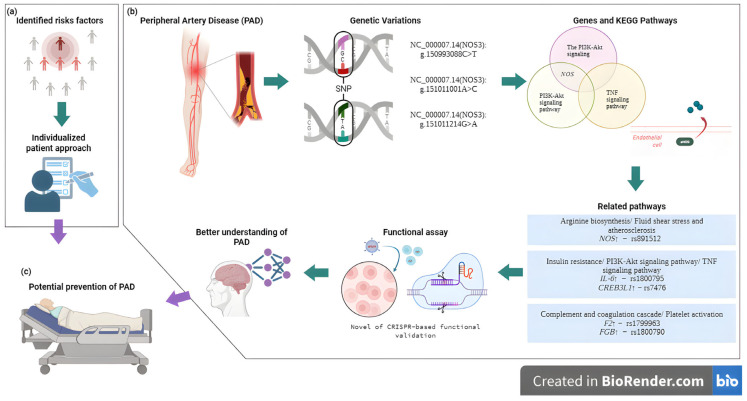
Overview of the integrative approach connecting polymorphisms, functional assays, and molecular pathways in PAD, providing insights into disease mechanisms and guiding potential preventive and therapeutic approaches. (**a**) Medical assessment and identification of individual risk factors for PAD. (**b**) Integration of genetic variation analysis and functional assays to elucidate key genes and pathways involved in PAD pathogenesis. (**c**) A future application of findings to personalized strategies for PAD prevention [105]. Created in BioRender. Villamil C (2025). https://app.biorender.com/illustrations/6904eab05a145cdd59b382d8 (accessed on 30 May 2025).

**Table 1 biomedicines-13-02723-t001:** Genetic variants identified in PAD.

Gene	SNP	Variant *	Role in PAD
Plaque origination			
*NOS3*	rs3918226	NC_000007.14(NOS3):g.150993088C>T	Variants in the *NOS3* gene modulate gene expression and enzymatic activity, ultimately affecting NO production, disrupting vascular tone regulation, and contributing to cardiovascular dysfunction. These variants influence *NOS3* at multiple levels. The rs3918226, located in the promoter region, reduces its transcriptional activity, resulting in reduced *NOS3* expression and diminished NO bioavailability. In contrast, the rs891512 variant potentially enhances transcription; however, this increased expression, under pro-oxidative conditions, might produce nitrosative stress and endothelial damage. The rs1808593 missense mutation results in a functional change in NOS3, decreasing its enzymatic activity and further limiting NO synthesis [25,27,28].
	rs891512	NC_000007.14(NOS3):g.151011001A>C	
	rs1808593	NC_000007.14(NOS3):g.151011214G>A	
*ICAM1*	rs5498	NC_000019.10(ICAM1):g.10285007A>G	This variant alters splicing and increases the expression of ICAM-1, enhancing leukocyte adhesion and endothelial activation. The *G* allele has been linked to increased levels of the soluble form of ICAM-1, promoting inflammation and plaque formation [27,28].
*SELE*	rs5361	NC_000001.11(SELE):g.169731919T>A	This variant is a missense mutation that alters the EGF-like domain of E-selectin, impacting leukocyte binding. This change contributes to increased endothelial activation, promoting vascular dysfunction and atheroma plaque formation [30].
*LIPC*	rs2070895	NC_000015.10(LIPC):g.58431740G>A	Variants such as rs2070895 and rs1800588, located in the promoter region of the *LIPC* gene, affect transcriptional regulation by altering binding sites, resulting in reduced hepatic lipase expression. This leads to higher LDL levels and lower HDL concentrations, ultimately impacting vascular health [30].
	rs1800588	NC_000015.10(LIPC):g.58431476C>A	
*LDLR*	rs138294113	NC_000019.10(LDLR): g.11081053C>T	This variant causes a nonsense change in the *LDLR* gene, leading to a truncated, non-functional LDL receptor or degradation of the mutant mRNA through nonsense-mediated decay. As a result, receptor availability on the cell surface decreases, impairing LDL clearance and promoting the formation of atherosclerotic plaques [7].
*LPA*	rs118039278	NC_000006.12(LPA):g.160564494G>A	Variants in the *LPA* gene, including rs118039278, rs3798220, rs10455872, and rs7452960, affect regulatory and coding regions, leading to altered mRNA splicing, transcript stability, or amino acid changes in apolipoprotein(a). These changes enhance the production or alter the structure of Lp(a), contributing to its accumulation and reduced clearance. This overexpression of *LPA* disrupts vascular homeostasis and facilitates atherosclerotic plaque development [38,52,53].
	rs3798220	NC_000006.12(LPA):g.160540105T>C	
	rs10455872	NC_000006.12:g.160589086A>G	
	rs7452960	NC_000006.12(LPA):g.160520609G>A	
*LPL*	rs322	NC_000008.11(LPL):g.19961706A>C	Variants of this gene affect the activity of lipoprotein lipase, potentially leading to altered triglyceride levels favoring their accumulation in the vascular system. The rs322 variant is located within an intronic region, which may influence *LPL* gene expression or splicing, thus modulating enzyme availability and contributing to lipid dysregulation [19,35].
*ABO*	rs505922	NC_000009.12(ABO):g.133273813C>T	The rs505922 variant in the *ABO* gene is found within a noncoding region and plays a regulatory role that may influence the expression of nearby genes involved in vascular biology. This polymorphism has been linked to changes in circulating levels of pro-inflammatory and pro-thrombotic factors, providing a possible genetic basis for the heightened cardiovascular risk observed in individuals with certain blood groups [7].
Plaque progression			
*IL-6*	rs1800795	NC_000007.13(IL6):g.22766645C>G	Polymorphisms of this gene are linked to elevated IL-6 in plasma, indicating high inflammatory activity. Among them, rs1800795, located in the promoter region, modulates IL-6 transcriptional activity. The *G* allele is associated with increased promoter activity and high IL-6 expression, enhancing vascular inflammation and atherogenesis. The variant rs4722172 is located in an intronic region and may influence IL-6 expression through regulatory mechanisms [7,30].
	rs4722172	NC_000007.13(IL6): g.22786532G>A	
*SH2B3*	rs653178 rs3184504	NC_000012.12:g.111569952C>A NC_000012.12:g.111446804T>A	The variant rs653178 in the *ATXN2-SH2B3* locus is significantly associated with an increased risk of PAD independent of traditional cardiovascular factors. It is in near-complete linkage disequilibrium with the missense variant rs3184504 in *SH2B3*, which affects immune and inflammatory pathways critical for vascular health. This variant influences endothelial function and inflammation, contributing to PAD development and related cardiovascular conditions such as myocardial infarction [54].
*CELSR2*	rs7528419	NC_000012.12:g.111446804T>A	Variants of this gene cause elevated levels of lipoprotein-associated phospholipase A2, contributing to the formation of atherosclerotic plaques, leading to arterial narrowing and reduced blood flow. The variant rs7528419 is in strong linkage disequilibrium with regulatory variants that enhance the hepatic expression of *SORT1*. This upregulation alters LDL metabolism, promoting lipid accumulation, vascular inflammation, and atherogenesis [7].
*HDAC9*	rs2107595	NC_000007.13(HDAC9): g.19049388G>A	This gene encodes histone deacetylase 9, an enzyme that removes acetyl groups from histones, leading to a more compact chromatin structure that represses gene transcription. This variant is located in a regulatory region and may alter transcription factor binding, indirectly modulating nearby genes and promoting vascular inflammation. In particular, the *A* allele is associated with the increased expression of *HDAC9*, which can enhance the transcriptional repression of anti-inflammatory genes while facilitating the expression of pro-inflammatory cytokines [7,44].
*CDKN2B-AS1*	rs1537372	NC_000009.11(CDKN2B-AS1):g.22103183G>A	Variants of this gene are associated with the increased expression of CDKN2B-AS1, leading to the enhanced repression of cell cycle regulatory genes. Variants such as rs1537372, located in an enhancer region, modulate transcription, while rs4977574 and rs785734 have been linked to altered chromatin states and increased CDKN2B-AS1 activity. These mechanisms facilitate vascular remodeling and promote a pro-inflammatory environment [7,38].
	rs4977574	NC_000009.11(CDKN2B-AS1):g.22098574A>G	
	rs7857345	NC_000009.11(CDKN2B-AS1):g.22087473T>A	
*PTPN11*	rs11066301	NC_000012.12(PTPN11):g.112433568A>G	This variant can influence alternative splicing or transcription efficiency, resulting in changes in PTPN11 levels. Altered PTPN11 expression impacts signaling cascades that regulate vascular cell proliferation and inflammatory responses, ultimately contributing to endothelial function and inflammation [7,38].
*CREB3L1*	rs7476	NC_000011.10(CREB3L1):g.46321284A>C	rs7476 is associated with the increased expression of *CREB3L1* by affecting mRNA stability and microRNA binding. This overexpression may influence lipid accumulation within vascular tissues, disrupt endothelial integrity, and promote vascular inflammation [7,38].
Plaque rupture			
*COL4A1*	rs1975514	NC_000013.11(COL4A1):g.110176544T>C	The rs1975514 variant may alter collagen stability, weakening the integrity of the vascular walls, and is an intronic variant located near a splice site in *COL4A1* that may influence mRNA processing or transcript stability, subtly affecting the production or quality of type IV collagen. This can compromise basement membrane integrity and increase vascular vulnerability [7].
*SMOC1*	rs55784307	NC_000014.9(SMOC1): g.70034647C>A	The polymorphism rs55784307, located in the genomic region of *SMOC1*, may influence matrix remodeling and cellular adhesion by altering regulation in gene expression. Such changes can disrupt the transcriptional control of *SMOC1*, modifying cellular responses to vascular injury and promoting a pro-inflammatory environment within the vascular wall [19].
*MMP3*	rs566125	NC_000011.10(MMP3):g.102839740C>T	This intronic variant may alter *MMP3* gene regulation, potentially enhancing enzyme expression. The resulting increase in the proteolytic activity of MMP3 can accelerate extracellular matrix degradation within the vascular wall, weakening plaque stability and increasing the risk of rupture and thrombosis [30,44].
*F5*	rs6025	NC_000001.11(F5):g.169549811C>A	The polymorphism rs6025 causes a single amino acid substitution in Factor V, making it resistant to cleavage by activated protein C. As a result, the anticoagulant pathway is impaired, promoting sustained thrombin generation and elevating the risk of abnormal clot formation [30,49].
*F2*	rs1799963	NC_000011.10(F2):g.46739505G>A	The rs1799963 variant can result in an altered prothrombin protein, often due to changes in the 3’ untranslated region of the gene, which may affect the regulation of gene expression. These changes can boost prothrombin production, leading to higher thrombin levels. This, in turn, disrupts the normal anticoagulation process, making abnormal blood clots more likely. On top of that, genetic variants of *F2* can also increase the expression of pro-inflammatory cytokines, which intensify inflammation in the blood vessels, speeding up the progression of PAD [50].
*FGB*	rs1800790	NC_000004.12:g.154562556G>A	The rs1800790 variant enhances fibrinogen beta gene expression, leading to elevated plasma fibrinogen levels. Increased fibrinogen promotes thrombosis and vascular inflammation, thereby increasing the risk of PAD. Studies such as the Edinburgh Artery Study have associated the *A* allele with higher fibrinogen concentrations and a greater risk of PA [49].

* Nomenclature according to ClinVar.

**Table 2 biomedicines-13-02723-t002:** Genetic variants replicated in multiple studies associated with PAD and their relationship with environmental and lifestyle factors, metabolic risks, and drugs.

Gene	Relevant Finding/ Clinic Relevance	Direction	Interaction with Environment/Lifestyle/Metabolic Risks/Drugs	SNP ID of Genetic Variants	Association with PAD	Reference
*NOS3*	Polymorphisms are associated with endothelial dysfunction and reduced NO bioavailability.	↓NO: ↑PAD risk	Yes: oxidative stress, diet, smoking, DM, hypertension.	rs891512 rs1808593 rs3918226	Contradictory results	[12,19,27,28,34,38,49,68,69,70,71]
*ICAM1*	There is an overexpression in endothelial cells and high circulating levels, contributing to the initiation and progression of atherosclerotic plaque.	↑ICAM1: ↑PAD risk	Yes: chronic inflammation, smoking, DM, and obesity.	rs5498 * rs5030352 *	Yes	[30,70,72,73,74,75,76]
*SELE*	Endothelial activation marker: its overexpression is associated with vascular dysfunction and atherosclerosis.	↑SELE: ↑PAD risk	Yes: smoking, inflammation, obesity, and insulin resistance.	rs5368 rs5356 rs5361	Yes	[29,70,73,77]
*LIPC*	Affects small LDL and HDL.	↑Activity: ↑sdLDL-C or ↓HDL and ↑PAD risk	Yes: diet, alcohol consumption, obesity, and exercise.	rs2070895 * rs1800588	Yes	[30,31,78,79,80]
*LDLR*	Genetic variants with loss of function increase LDL levels and promote accelerated atherosclerosis.	LDLR defective ↑LDL: ↑PAD risk	Yes: diet, smoking, alcohol consumption, and exercise.	rs651172 rs1122608 rs138294113 *	Yes	[7,19,30,33,49]
*LPA*	Elevated levels of Lp(a) are associated with acute atherothrombotic events, aortic stenosis, and PAD.	↑Lp(a): ↑PAD risk	Yes: diet, smoking, alcohol consumption, obesity, and statins.	rs10455872 * rs7452960 * rs3798220 rs118039278 *	Yes	[30,34,35,38,53,81,82,83,84]
*LPL*	Genetic variants that favor loss/gain of function increase/decrease triglycerides.	↓LDL function: ↑PAD risk ↑LDL function: ↓PAD risk	Yes: diet, alcohol consumption, exercise, and DM2.	rs328 rs322 *	Yes	[19,30,34]
*ABO*	Regulates levels of lipids, adhesion molecules, vWF, and FVIII; affects lipid metabolism, inflammation, and thrombosis. In non-O groups, a prothrombotic state is favored, and it is associated with the presence and severity of PAD.	↑vWF and FVIII: ↑Thrombosis risk	Yes: smoking, hypertension, and coagulation factors.	rs505922 * rs616154 * rs635634 rs8176719	Yes	[30,34,39,85]
*IL-6*	The increase in IL-6 is associated with increased inflammation.	↑IL-6: ↑PAD and adverse cardiovascular events risk	Yes: chronic inflammation, smoking, alcohol consumption, obesity, sedentary lifestyle, and DM2.	rs4722172 * rs1800795 rs2228145 rs2069827 *	Contradictory results	[12,30,41,70,73,76,77,86,87,88]
*SH2B3*	Participates in the signaling of immune and hematopoietic cells, promotes inflammation, vascular tone, and PAD.	↑ PAD risk with risk alleles	Yes: hypertension and obesity.	rs3184504 * rs7528419 rs653178	Yes	[7,12,30,38,41,42,54,85,89]
*CELSR2*	Alters LDL metabolism, promoting lipid accumulation, inflammation and atherosclerosis.	↑LDL: ↑Atherosclerosis risk	Yes: diet, dyslipidemia, and obesity.	rs12740374	No	[12,19,38,85]
*HDAC9*	Modulates inflammation and the VSMC phenotype and is associated with vascular calcification and atherosclerosis.	↑Expression/activity: ↑PAD risk	Yes: smoking, hypertension, and DM2.	rs2107595 rs2074633 *	Yes	[7,30,38,49,68]
*CDKN2B-AS1*	Regulates the cell cycle, promotes inflammation and VSMC proliferation.	↑CDKN2B-AS1 expression: ↑Atherosclerosis and PAD risk	Yes: smoking and DM2.	rs1537372 * rs10738610 * rs1333049 rs10757278 rs10757269	Yes	[30,34,38,49,52,54,90]
*PTPN11*	Participates in endothelial signaling, promotes inflammation, and is associated with pro-atherogenesis	↑Function: ↑PAD risk	Yes: hypertension and DM2.	rs11066301 * rs10774624	Yes	[12,19,30,34,38]
*CREB3L1*	Modulates the stress response, is associated with vascular calcification, and promotes cellular remodeling and inflammation.	Depends on the context	Yes: obesity and DM2.	rs7476 *	Contradictory results	[7,19,30,38,48]
*COL4A1*	Participates in the integrity of the basement membrane; its genetic variants cause microvascular fragility.	Defect: ↑Fragility	Yes: smoking and hypertension.	rs1975514 *	Contradictory results	[7,12,30,36,91]
*SMOC1*	Regulates cellular homeostasis, cell migration and proliferation, calcification, tissue fibrosis, and angiogenesis.	↑SMOC1: ↑Calcification/cell remodeling	Yes: DM2.	rs55784307 *	Contradictory results	[12,30,34,38,47]
*MMP-3*	Overexpression is associated with extracellular matrix degradation and vulnerability of the atherosclerotic plaque.	↑MMP3: Instability/rupture of the atherosclerotic plaque	Yes: smoking, obesity, and DM2.	rs566125 * rs5030352	Yes	[7,30,48,70,77]
*F5*	The Factor V Leiden mutation causes resistance to protein C, which promotes a prothrombotic state in PAD.	↑Hypercoagulability	Yes: smoking, obesity, pregnancy, and contraceptives.	rs6025 *	Contradictory results	[7,12,19,30,49,50,73,92]
*F2*	Mutations in prothrombin increase the risk of clots forming on the atherosclerotic plaque.	↑Activity: ↑Thrombotic risk	Yes: smoking, obesity, pregnancy, and contraceptives.	rs1799963	Contradictory results	[50,73,92,93,94,95]
*FGB*	Overexpression is associated with elevated plasma fibrinogen, inflammation, thrombosis and worse prognosis.	↑Expression: ↑Fibrinogen levels and this, ↑PAD and CVDs risk	Yes: smoking, alcohol consumption, inflammation, obesity, and DM2.	rs4220 rs1800790	Contradictory results	[5,49,51,96,97,98]

* Genetic variants have been replicated in multiple populations or meta-analyses. ABO: alpha 1-3-*N*-acetylgalactosaminyltransferase and alpha 1-3-galactosyltransferase; CDKN2B: Cyclin-dependent kinase inhibitor 2B-antisense 1; CELSR2: Cadherin EGF LAG seven-Pass G-type receptor 2; COL4A1: Collagen type IV alpha 1 chain; CREB3L1: CAMP responsive element binding protein 3 like 1; CVDs: Cardiovascular diseases; DM2: Type 2 diabetes mellitus; F2: Factor 2 (Prothrombin); F5: Factor 5; FGB: Fibrinogen beta chain; FVIII: Factor VIII; HDAC9: Histone deacetylase 9; HDL: High-density lipoprotein; ICAM1: Intercellular adhesion molecule 1; IL-6: Interleukin-6; LIPC: Lipase C, hepatic type; LDL: Low-density lipoprotein; LDLR: LDL receptor; Lp(a): Lipoprotein (a); LPL: Lipoprotein lipase; MMP3: Matrix metallopeptidase 3; NOS3: Nitric oxide synthase 3; PTPN11: Protein tyrosine phosphatase non-receptor type 11; sdLDL-C: Small dense low-density lipoprotein cholesterol; SELE: Selectin E; SH2B3: SH2B adaptor protein 3; VSMC: Vascular smooth muscle cell; vWF: von Willebrand factor; ↓ Down regulated; ↑ Up regulated

**Table 3 biomedicines-13-02723-t003:** Factors that influence the failure to replicate findings in genetic association studies.

Factor	How the Result Could Be Affected	How To Avoid It	Factor	Reference
Sample size	Published research demonstrates substantial variability in sample sizes across PAD studies. In many cases, sample sizes were limited and determined by convenience, thereby restricting the statistical power to detect significant associations.	Use power analysis before starting the study (based on expected variant frequency, estimated OR, alpha). Ensure that the sample has ≥80% power to detect significant associations.	Sample size	Low (<100) [10,13,26,28] High (>500) [22,38,54,99]
Heterogeneity in diagnostic criteria/Definition of PAD	The diagnostic criteria for PAD varied considerably across studies. In some cohorts, diagnosis was established solely by ABI, claudication symptoms, history of revascularization, chronic ischemia, or non-traumatic amputation. In others, diagnosis relied on a more comprehensive assessment that combined medical history, detailed physical examination, and confirmatory tests such as imaging. This heterogeneity in case definition may influence both the magnitude and the direction of the reported associations.	Use consensus definitions from international guidelines (e.g., ABI ≤ 0.90 for PAD, CLTI criteria from the Global Vascular Guidelines). Avoid diagnoses based solely on nonspecific symptoms or poorly validated medical records. Include symptoms and procedures (claudication, revascularization, amputation) only as additional criteria. Validate and, if possible, integrate emerging tools (e.g., PAT) in patients where the ABI may be unreliable.	Heterogeneity in diagnostic criteria/Definition of PAD	[1,2,3,6,99]
Study of PAD alone or associated with classic cardiovascular risk factors or environmental factors	Some studies focused exclusively on patients diagnosed with PAD, whereas others included individuals with PAD in combination with at least one traditional cardiovascular risk factor, such as DM2 or hypertension. This variability in inclusion criteria may introduce heterogeneity and affect the comparability of results across studies.	Analysis stratification (conduct separate analyses for PAD alone and for PAD with comorbidities). Include classic risk factors as covariates to reduce bias.	Study of PAD alone or associated with classic cardiovascular risk factors or environmental factors	[99,100]
Type of analysis/study power	Some studies performed analyses adjusting for potential confounders, including hypertension, DM2, sex, age, and smoking, and in some cases incorporated interaction models. In contrast, others did not, which may obscure true effects within specific subgroups. Furthermore, several studies reported statistical power below 50%, limiting the reliability of their findings.	Perform a stratified design and analysis (e.g., severity, ancestry), harmonizing data, and, use matching and predefined interaction models. Increase the number of cases by working with cohorts from different centers.	Type of analysis/study power	[14,15]
Study design	Population selection varied across studies and included differences in outpatient versus hospitalized cohorts, as well as demographic and socioeconomic characteristics such as age, sex, ancestry, ethnicity, and socioeconomic status. These factors may significantly influence disease prevalence and outcomes, thereby affecting comparability across studies.	Clearly define the research question, hypothesis, and objectives (use the PICO/PECO framework). Standardize the definition of variables.	Study design	[68,101,102]
Standardization of the technique and determinations	Rigorous standardization and the meticulous execution of laboratory measurements are essential to ensure accuracy, reproducibility, and comparability across studies. Inadequate standardization can introduce systematic error, bias results, and hinder the validity of cross-study comparisons.	Adhere to the CLSI and ISO 15189:2022 guidelines. Ensure the training and certification of operational personnel.	Standardization of the technique and determinations	[103]

ABI: ankle–brachial index; CLSI: Clinical and Laboratory Standards Institute; DM2: type 2 diabetes mellitus; ISO: The International Organization for Standardization; PAD: peripheral artery disease; PAT: Plantar Acceleration Time; PICO acronym: P = Population/Problem, I = Intervention, C = Comparison/Control, O = Outcome; PECO acronym: P = Population/Problem, E = Exposure, C = Comparison, O = Outcome.

## Data Availability

No new data were created or analyzed in this study. Data sharing is not applicable to this article.

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
