# Peer review of "Genetic Insights into Peripheral Artery Disease: A Narrative Review"

_biomedicines, 2025, doi:10.3390/biomedicines13112723_

Round 1

Reviewer 1 Report

Comments and Suggestions for Authors

The present manuscript is a review that investigates general genetic factors regarding Peripheral Artery Disease (PAD), the focus of the paper is to establish some of the main genetic factors that interfere with the onset and progression of the disease. The general organization is acceptable and the reference are up-to-date. On the other hand, the review shows certain aspects that hinders the reader’s comprehension affecting academic rigor.

  1. Strengths of the manuscript

PAD is a relevant scientific hot topic regarding public health and the genetic cause underlying the disease. Research and health care require comprehensive and scalable methods for the discovery of novel disease targets and evolutionary drivers on PAD, and the authors provide a detailed description of genes and variants associated with the disease. The text is well-organized into groups of gene function on the pathophysiology of PAD. The table and the figure presented by the authors are valid summarizing tools.

  1. Methodological Aspects of the manuscript

The paper is a narrative review. The authors claim to provide a "review" but they do not specify the search methodology used, such as the inclusion and the exclusion criteria, nor the quality evaluation of included studies. Hence, I suggest explicitly stating whether this is a narrative or scoping review and how studies were selected, including information regarding databases searched, keywords, and other descriptors.

Moreover, the authors acknowledge possible bias in selection, which weakens the scientific basis of the review. Lack of critical appraisal of sources is a concern, so a careful and systematic assessment of the paper’s trustworthiness could improve substantially its methodological rigor.

  1. In-depth analysis required to improve the manuscript

The authors collect data on PAD-associated genes and their variants (such as HDAC9, SH2B3, MMP3), but I felt like the manuscript needs a deeper mechanistic discussion on those aspects. Most paragraphs regarding that topic are presented as standalone associations rather than woven into broader biological pathways. The authors should consider to link the description of a gene or variant to the onset and progression of the disease (inflammation, lipid metabolism, thrombosis, among others). This would improve the understand of the overall genetic role in PAD.

Additionally, I think that a critical synthesis of the scientific literature is rather missing. The scientific value of the paper would be greatly increased by filling in these kinds of gaps. A) Which results are reliably repeated in different studies? What could be the cause of the disparities and where do they exist? Addressing these kinds of gaps would also significantly increase the paper’s scientific value.

  1. Lack of Clinical Integration

The study ought to focus more on the real implications of these genetic findings for actual health care or patients. Do gene variations interefere in the early disease diagnosis, treatment prognostics, or risk factor prediction? Toss in something concrete about how these discoveries could shape diagnosis, prognosis, or therapy.

  1. Redundancy and Structure of the manuscript

Repetition and Structure - The intro and conclusion keep focusing on about how PAD is a big deal worldwide. Condense the repeating bits in other to make the text more clarified and objective.

  1. Abstract

Abstract The abstract ends abruptly: "we summarized an overview of the genetic factors associated with peripheral artery disease." This is vague. I suggest that the abstract should be revised to include the main findings, conclusions, methods and scope of the review.

  1. References

Most references are up-to-date; however, some citations are inconsistently formatted (some should be cited more than once but without a difference in numbering) and some listings are repeated (e.g., [8] and [32] appear too often with no distinction). Format consistently and diversify sources. 

  1. Questions

Please take a moment to address the questions listed below in order to increase the scientific value of the paper. I guess if you could link genes and their variants to metabolic pathways (you could you KEGG database for example) and try and hypothesize (supported by the literature, of course) how they could be related to the onset and progression of the disease, the authors then could enhance the scientific validity of the manuscript and our understanding of PAD genetics. Moreover, there are some aspects that could be deeper explored, for example, the gene-environment interactions (habits, smoking, exercise), ethnic variations (point to data from specific populations), emerging technologies (epigenetics or functional genomics) and so many others that could elevate the robustness of the review, paving the way to be used as an authoritative resource for health care practitioners and researchers. As I read the paper, the following are some of the questions that crossed my mind:

  1. How do the genetic variants described converge on common pathophysiological pathways?
  2. Which of these genetic variants have been replicated in multiple populations or meta-analyses?
  3. How can these genetic discoveries be applied clinically?
  4. How genetic features of PAD differ from the other classical atherosclerotic diseases such as CAD or stroke?
  5. Have any of these genes or variants shown interactions with environmental or lifestyle risk factors (e.g., smoking, diabetes, hypertension)?
  6. I think that it would enrich the discussion section of the manuscript if the authors could add some aspects regarding new technology such as single-cell transcriptomics or CRISPR-based functional validation in order to enhance further our general understanding of PAD genetics.

  1. I felt the manuscript lacking figures and diagrams. Additional figures (such as a schematic overview of gene-pathway relationships in PAD, or a translational flowchart linking genetics to clinical application) would greatly enhance the accessibility, pedagogical value and robustness of the review. Considering the genetic-density of the data presented, smart visuals could help readers power through this Gordian molecular knot to better comprehend the links between the molecular mechanisms and the disease outcomes.

Suggested Additions:

Figure 2: A visual map of key genes and their roles across the stages of PAD (origination → progression → rupture).

Figure 3: A translational roadmap showing how identified genes/SNPs could contribute to risk prediction, therapeutic targets, or biomarker development.

Comments on the Quality of English Language

Title and Language Quality

Title: “Genetic Basis into Peripheral Artery Disease” is grammatically incorrect. Suggestion:
→ "Genetic Basis of Peripheral Artery Disease" or
→ "Genetic Insights into Peripheral Artery Disease"

Language & Grammar: The manuscript contains frequent grammatical issues, awkward constructions, and unnatural phrasing.

Example: “This considering, that ABI it is a noninvasive test…” → should be corrected to a more fluent form.

Repeated use of "it is" or “this is” disrupts clarity.

Action: A professional English language editing is essential before publication.

Author Response

REVIEWER 1

We thank to the reviewer for the insightful comments and helpful criticisms. We have responded to all comments as detailed below and we now hope that you will find our revised Manuscript acceptable for publication in “Biomedicines”.

Comments and Suggestions for Authors

The present manuscript is a review that investigates general genetic factors regarding Peripheral Artery Disease (PAD), the focus of the paper is to establish some of the main genetic factors that interfere with the onset and progression of the disease. The general organization is acceptable and the reference are up-to-date.

  1. Strengths of the manuscript

PAD is a relevant scientific hot topic regarding public health and the genetic cause underlying the disease. Research and health care require comprehensive and scalable methods for the discovery of novel disease targets and evolutionary drivers on PAD, and the authors provide a detailed description of genes and variants associated with the disease. The text is well-organized into groups of gene function on the pathophysiology of PAD. The table and the figure presented by the authors are valid summarizing tools.

  1. Methodological Aspects of the manuscript

The paper is a narrative review. The authors claim to provide a "review" but they do not specify the search methodology used, such as the inclusion and the exclusion criteria, nor the quality evaluation of included studies. Hence, I suggest explicitly stating whether this is a narrative or scoping review.

ANSWER:

We agree with the reviewer; it is a narrative review. We took into account your suggestions in the manuscript title.

In order to clarify this issue, we adapted the title of the manuscript “Genetic Insights into Peripheral Artery Disease: A narrative review” (Line 2 - 3).

  1. In-depth analysis required to improve the manuscript

The authors collect data on PAD-associated genes and their variants (such as HDAC9, SH2B3, MMP3), but I felt like the manuscript needs a deeper mechanistic discussion on those aspects. Most paragraphs regarding that topic are presented as standalone associations rather than woven into broader biological pathways. The authors should consider to link the description of a gene or variant to the onset and progression of the disease (inflammation, lipid metabolism, thrombosis, among others). This would improve the understand of the overall genetic role in PAD.

ANSWER:

Thank you to the reviewer. We have taken your recommendation into account and it is shown the answer in question 1a.

Additionally, I think that a critical synthesis of the scientific literature is rather missing. The scientific value of the paper would be greatly increased by filling in these kinds of gaps. A) Which results are reliably repeated in different studies? What could be the cause of the disparities and where do they exist? Addressing these kinds of gaps would also significantly increase the paper’s scientific value.

ANSWER:

We are in accordance with the reviewer. We have taken your recommendation into account and it is shown in detail in Table 2 (question 2a, 5a) and Table 3 we described the possible cause of the disparities (LINES 543-545).

  1. Lack of Clinical Integration

The study ought to focus more on the real implications of these genetic findings for actual health care or patients. Do gene variations interfere in the early disease diagnosis, treatment prognostics, or risk factor prediction? Toss in something concrete about how these discoveries could shape diagnosis, prognosis, or therapy.

ANSWER:

Thank you to the reviewer. We have taken your recommendation into account and it is shown the answer in question 3a.

  1. Redundancy and Structure of the manuscript

Repetition and Structure - The intro and conclusion keep focusing on about how PAD is a big deal worldwide. Condense the repeating bits in other to make the text more clarified and objective.

ANSWER:

Thank you to the reviewer. We have taken your recommendation into account.

  1. Abstract

Abstract The abstract ends abruptly: "we summarized an overview of the genetic factors associated with peripheral artery disease." This is vague. I suggest that the abstract should be revised to include the main findings, conclusions, methods and scope of the review.

ANSWER:

Thank you for your comment. We have now improved the abstract based on a narrative review.

The abstract is described in LINES 22-32.

  1. References

Most references are up-to-date; however, some citations are inconsistently formatted (some should be cited more than once but without a difference in numbering) and some listings are repeated (e.g., [8] and [32] appear too often with no distinction). Format consistently and diversify sources.

ANSWER:

Thank you to the reviewer for this important recommendation. in accordance with the reviewer, we have addressed this situation.

SPECIFIC QUESTIONS:

1a. How do the genetic variants described converge on common pathophysiological pathways?

ANSWER:

We appreciate the reviewer’s insightful suggestion. To address this point, we expanded the manuscript to include molecular mechanisms of the identified genes.

To further contextualize these findings, the PAD-associated genes involved in the origin, progression, and rupture of the plaque were scrutinized through ExpressAnalyst (web-based platform) for functional enrichment analysis to KEGG pathways, and the overlap of genes involved in each pathway is highlighted. (LINES 428-477).

To highlight the relevance of the genes described in Table 1 associated with PAD, a KEGG pathway enrichment analysis was performed to represent the pathways involved in these genes' disease processes. In this regard, at the plaque formation stage, it allowed for the integration of the arginine biosynthesis pathway, the glycosphingolipid pathway, and the fluid shear stress and atherosclerosis pathway. Genes such as NOS3, ABO, and SELE are involved in these pathways (Figure 2A).

The importance of these pathways is as follows:

1) Arginine biosynthesis produces NO, a molecule that relaxes and dilates blood vessels, improving blood flow. In addition, L-arginine may improve walking ability and blood flow in these patients [54,55].

2) The biosynthesis of glycosphingolipids and their metabolites, when altered, can lead to their accumulation in tissues. This accumulation has been linked to the development and progression of PAD. It is suggested that alterations in sphingolipid metabolism contribute to the cellular and tissue damage that occurs during the atherosclerotic process [56].

3) Fluid shear stress and atherosclerosis, due to the force exerted by the constant flow of blood on the walls of blood vessels, play an important role in atherogenesis by altering the integrity of the endothelium, increasing its permeability, and allowing the entry of lipoproteins and inflammatory cells, which initiates the process of plaque formation and thrombi in the arteries and causes PAD [57].

Moreover, the pathways related to the stage of plaque progression include the PI3K-Akt signaling pathway, TNF signaling pathway, and insulin resistance pathway. Genes such as NOS3, IL-6, CREB3L1, PTPN11, and SELE are involved in these pathways (Figure 2B). The relevance of these pathways is as follows:

1) The PI3K-Akt signaling pathway is involved in regulating inflammation and oxidative stress, both of which are key factors in the pathogenesis of PAD [58].

2) The TNF signaling pathway promotes oxidative stress and decreases the bioavailability of NO, a crucial vasodilator, contributing to endothelial dysfunction. Furthermore, the TNF signaling pathway impacts vascular remodeling processes, leading to structural changes in the arteries of patients [59].

3) The insulin resistance pathway causes vascular damage through endothelial dysfunction via the inhibition of NO production. In addition, insulin resistance can activate pro-inflammatory molecular pathways, such as the MAP kinase (MAPK) pathway, which contribute to the disease [60].

Finally, the pathways related to the plaque rupture stage allow for the integration of complement activation, coagulation cascade, and platelet activation, which are involved in exacerbating the inflammatory process, which, together with endothelial dysfunction promotes thrombosis. Genes such as F5, F2, and FGB are involved in these pathways (Figure 2C) [61,62].

(a) Plaque origination

(b) Plaque progression

(c) Plaque rupture

Figure 2. PAD genes and their connection to KEGG pathways. The PAD-associated genes involved in the origin, progression, and rupture of plaque were scrutinized using ExpressAnalyst (a web-based platform) for functional enrichment analysis to KEGG pathways. The overlap of genes involved in each pathway is highlighted [53]. (a) KEGG pathways interaction of plaque origination. (b) KEGG pathways interaction of plaque progression. (c) KEGG pathways interaction of plaque rupture.

2a. Which of these genetic variants have been replicated in multiple populations or meta-analyses?

 ANSWER:

In accordance with the reviewer, we added in Table 2, this important information in the discussion section (LINES 516-526).

Table 2. Genetic variants replicated in multiple studies associated with PAD and their relation with environmental and lifestyle factors, metabolic risks, and drugs.

Gene

Relevant finding/ Clinic relevance

Direction

Interaction with environment/ lifestyle/metabolic risks/drugs

SNP ID of genetic variants

Association with PAD

Reference

NOS3

Polymorphisms are associated with endothelial dysfunction and reduced NO bioavailability.

↓NO: ↑PAD risk

Yes: oxidative stress, diet, smoking, DM, hypertension.

rs891512

rs1808593

rs3918226

Contradictory results

[12,19,26,27,33,37,48,66–69]

ICAM1

There is an overexpression in endothelial cells and high circulating levels, contributing to the initiation and progression of atherosclerotic plaque.

↑ICAM1: ↑PAD risk

Yes: chronic inflammation, smoking, DM, and obesity.

rs5498*

rs5030352*

Yes

[29,68,70–74]

SELE

Endothelial activation marker: its overexpression is associated with vascular dysfunction and atherosclerosis.

↑SELE: ↑PAD risk

Yes: smoking, inflammation, obesity, and insulin resistance.

rs5368

rs5356

rs5361

Yes

[28,68,71,75]

LIPC

Affects small LDL and HDL.

↑Activity: ↑sdLDL-C or ↓HDL and ↑PAD risk

Yes: diet, alcohol consumption, obesity, and exercise.

rs2070895*

rs1800588

Yes

[29,30,76–78]

LDLR

Genetic variants with loss of function increase LDL levels and promote accelerated atherosclerosis.

LDLR defective ↑LDL: ↑PAD risk

Yes: diet, smoking, alcohol consumption, and exercise.

rs651172

rs1122608

rs138294113*

Yes

[7,19,29,32,48]

LPA

Elevated levels of Lp(a) are associated with acute atherothrombotic events, aortic stenosis, and PAD.

↑Lp(a): ↑PAD risk

Yes: diet, smoking, alcohol consumption, obesity, and statins.

rs10455872*

rs7452960*

rs3798220

rs118039278*

Yes

[29,33,34,37,51,79–82]

LPL

Genetic variants that favor loss/gain of function increase/decrease triglycerides.

↓LDL function: ↑PAD risk

↑LDL function: ↓PAD risk

Yes: diet, alcohol consumption, exercise, and DM2.

rs328

rs322*

Yes

[19,29,33]

ABO

Regulates levels of lipids, adhesion molecules, vWF, and FVIII; affects lipid metabolism, inflammation, and thrombosis. In non-O groups, a prothrombotic state is favored, and it is associated with the presence and severity of PAD.

↑vWF and FVIII: ↑Thrombosis risk

Yes: smoking, hypertension, and coagulation factors.

rs505922*

rs616154*

rs635634

rs8176719

Yes

[29,33,38,83]

IL-6

The increase in IL-6 is associated with increased inflammation.

↑IL-6: ↑PAD and adverse cardiovascular events  risk

Yes: chronic inflammation, smoking, alcohol consumption, obesity, sedentary lifestyle, and DM2.

rs4722172*

rs1800795

rs2228145

rs2069827*

Contradictory results

[12,29,40,68,71,74,75,84–86]

SH2B3

Participates in the signaling of immune and hematopoietic cells, promotes inflammation, vascular tone, and PAD.

↑ PAD risk with risk alleles

Yes: hypertension and obesity.

rs3184504*

rs7528419

rs653178

Yes

[7,12,29,37,40,41,52,83,87]

CELSR2

Alters LDL metabolism, promoting lipid accumulation, inflammation and  atherosclerosis.

↑LDL: ↑Atherosclerosis risk

Yes: diet, dyslipidemia, and obesity.

rs12740374

No

[12,19,37,83]

HDAC9

Modulates inflammation and the VSMC phenotype and is associated with vascular calcification and atherosclerosis.

↑Expression/activity: ↑PAD risk

Yes: smoking, hypertension, and DM2.

rs2107595

rs2074633*

Yes

[7,29,37,48,66]

CDKN2B-AS1

Regulates the cell cycle, promotes inflammation and VSMC proliferation.

↑CDKN2B-AS1 expression: ↑ Atherosclerosis and PAD risk

Yes: smoking and DM2.

rs1537372*

rs10738610*

rs1333049

rs10757278

rs10757269

Yes

[29,33,37,48,50,52,88]

PTPN11

Participates in endothelial signaling, promotes inflammation, and is associated with pro-atherogenesis

↑Function: ↑PAD risk

Yes: hypertension and DM2.

rs11066301*

rs10774624

Yes

[12,19,29,33,37]

CREB3L1

Modulates the stress response, is associated with vascular calcification, and promotes cellular remodeling and inflammation.

Depends on the context

Yes: obesity and DM2.

rs7476*

Contradictory results

[7,19,29,37,47]

COL4A1

Participates in the integrity of the basement membrane; its genetic variants cause microvascular fragility.

Defect:

↑Fragility

Yes: smoking and hypertension.

rs1975514*

Contradictory results

[7,12,29,35,89]

SMOC1

Regulates cellular homeostasis, cell migration and proliferation, calcification, tissue fibrosis, and angiogenesis.

↑SMOC1: ↑Calcification/ cell remodeling

Yes: DM2.

rs55784307*

Contradictory results

[12,29,33,37,46]

MMP-3

Overexpression is associated with extracellular matrix degradation and vulnerability of the atherosclerotic plaque.

↑MMP3:

Instability/rupture of the atherosclerotic plaque

Yes: smoking, obesity, and DM2.

rs566125*

rs5030352

Yes

[7,29,47,68,75]

F5

The Factor V Leiden mutation causes resistance to protein C, which promotes a prothrombotic state in PAD.

↑Hypercoagulability

Yes: smoking, obesity, pregnancy, and contraceptives.

rs6025*

Contradictory results

[7,12,19,29,48,49,71,90]

F2

Mutations in prothrombin increase the risk of clots forming on the atherosclerotic plaque.

↑Activity: ↑Thrombotic risk

Yes: smoking, obesity, pregnancy, and contraceptives.

rs1799963

Contradictory results

[49,71,91–93]

FGB

Overexpression is associated with elevated plasma fibrinogen, inflammation, thrombosis and worse prognosis.

↑Expression: ↑Fibrinogen levels and this, ↑PAD and CVDs risk

Yes: smoking, alcohol consumption, inflammation, obesity, and DM2.

rs4220

rs1800790

Contradictory results

[5,48,94–97]

* Genetic variants have been replicated in multiple populations or meta-analyses. ABO: alpha 1-3-N-acetylgalactosaminyltransferase and alpha 1-3-galactosyltransferase; CDKN2B: Cyclin-dependent kinase inhibitor 2B-antisense 1; CELSR2: Cadherin EGF LAG seven-Pass G-type receptor 2; COL4A1: Collagen type IV alpha 1 chain; CREB3L1: CAMP responsive element binding protein 3 like 1; CVDs: Cardiovascular diseases; DM2: Type 2 diabetes mellitus; F2: Factor 2 (Prothrombin); F5: Factor 5; FGB: Fibrinogen beta chain; FVIII: Factor VIII; HDAC9: Histone deacetylase 9; HDL: High-density lipoprotein; ICAM1: Intercellular adhesion molecule 1; IL-6: Interleukin-6; LIPC: Lipase C, hepatic type; LDL: Low-density lipoprotein; LDLR: LDL receptor; Lp(a): Lipoprotein (a); LPL: Lipoprotein lipase; MMP3: Matrix metallopeptidase 3; NOS3: Nitric oxide synthase 3; PTPN11: Protein tyrosine phosphatase non-receptor type 11; sdLDL-C: Small dense low-density lipoprotein cholesterol; SELE: Selectin E; SH2B3: SH2B adaptor protein 3; VSMCs: Vascular smooth muscle cells; vWF: von Willebrand factor.

3a. How can these genetic discoveries be applied clinically?

ANSWER:

Thanks to the reviewer. Now, we added this information in the perspectives section for this important point and we included the figure 3 to represents this issue (558- 575).

The integration of genetic insights into PAD management has important clinical implications, particularly for risk stratification and personalized prevention. A proposed clinical pathway begins with the identification of at-risk individuals, combining family history and conventional factors with polygenic risk scores when available [103]. In those at elevated genetic risk, early non-invasive vascular screening (ankle–brachial index, duplex ultrasound) may be warranted, followed by intensified preventive strategies, including optimized lipid, blood pressure, and glycemic control, as well as structured smoking cessation. Genetic insights may also guide patient selection for novel therapies or clinical trials targeting PAD-specific molecular mechanisms. This stepwise integration of genetic risk into vascular practice exemplifies the potential of precision medicine to improve outcomes in PAD [12]. (Figure 3).

Figure 3. Overview of the integrative approach connecting polymorphisms, functional assays, and molecular pathways in PAD, providing insights into disease mechanisms and guiding potential preventive and therapeutic approaches. (a) Medical assessment and identification of individual risk factors for PAD. (b) Integration of genetic variation analysis and functional assays to elucidate key genes and pathways involved in PAD pathogenesis. (c) A future application of findings to personalized strategies for PAD prevention. Created with Biorender.

4a. How genetic features of PAD differ from the other classical atherosclerotic diseases such as CAD or stroke?

ANSWER:

We acknowledge to the reviewer for this important distinction. To identify PAD-specific patterns beyond general atherosclerosis, several strategies have been suggested:

In order to clarify this important point, we added the following paragraph (LINES 118 - 132).

On the other hand, it is important to differentiate the genetic pattern for PAD rather than atherosclerosis. Recent studies emphasize the importance of differentiating PAD-specific genetic signatures from those driving coronary or cerebrovascular disease. Several methodological approaches may help delineate this specificity, including case-only analyses in PAD patients without CAD or stroke, conditional GWAS adjusting for shared atherosclerotic risk, and tissue-specific expression studies in peripheral arteries [19]. Transcriptomic and expression quantitative trait locus analyses suggest that certain variants influence gene expression in limb vasculature more strongly than in coronary or cerebral beds, pointing to localized mechanisms such as impaired angiogenesis and collateral formation [12]. Importantly, PAD progression is shaped by unique biomechanical and hemodynamic factors in peripheral circulation, such as shear stress, collateral vessel development, and ischemia–reperfusion dynamics, which may account for functional differences in the genetic determinants of disease expression. Thus, PAD genetics are characterized by both shared systemic pathways and modifiers with context-specific effects in the peripheral vasculature.

5a. Have any of these genes or variants shown interactions with environmental or lifestyle risk factors (e.g., smoking, diabetes, hypertension)?

ANSWER:

Thanks to the reviewer for this suggestion. Now, we included in Table 2 this information in the discussion section (LINES 516 – 526).

Table 2. Genetic variants replicated in multiple studies associated with PAD and their relation with environmental and lifestyle factors, metabolic risks, and drugs.

Gene

Relevant finding/ Clinic relevance

Direction

Interaction with environment/ lifestyle/metabolic risks/drugs

SNP ID of genetic variants

Association with PAD

Reference

NOS3

Polymorphisms are associated with endothelial dysfunction and reduced NO bioavailability.

↓NO: ↑PAD risk

Yes: oxidative stress, diet, smoking, DM, hypertension.

rs891512

rs1808593

rs3918226

Contradictory results

[12,19,26,27,33,37,48,66–69]

ICAM1

There is an overexpression in endothelial cells and high circulating levels, contributing to the initiation and progression of atherosclerotic plaque.

↑ICAM1: ↑PAD risk

Yes: chronic inflammation, smoking, DM, and obesity.

rs5498*

rs5030352*

Yes

[29,68,70–74]

SELE

Endothelial activation marker: its overexpression is associated with vascular dysfunction and atherosclerosis.

↑SELE: ↑PAD risk

Yes: smoking, inflammation, obesity, and insulin resistance.

rs5368

rs5356

rs5361

Yes

[28,68,71,75]

LIPC

Affects small LDL and HDL.

↑Activity: ↑sdLDL-C or ↓HDL and ↑PAD risk

Yes: diet, alcohol consumption, obesity, and exercise.

rs2070895*

rs1800588

Yes

[29,30,76–78]

LDLR

Genetic variants with loss of function increase LDL levels and promote accelerated atherosclerosis.

LDLR defective ↑LDL: ↑PAD risk

Yes: diet, smoking, alcohol consumption, and exercise.

rs651172

rs1122608

rs138294113*

Yes

[7,19,29,32,48]

LPA

Elevated levels of Lp(a) are associated with acute atherothrombotic events, aortic stenosis, and PAD.

↑Lp(a): ↑PAD risk

Yes: diet, smoking, alcohol consumption, obesity, and statins.

rs10455872*

rs7452960*

rs3798220

rs118039278*

Yes

[29,33,34,37,51,79–82]

LPL

Genetic variants that favor loss/gain of function increase/decrease triglycerides.

↓LDL function: ↑PAD risk

↑LDL function: ↓PAD risk

Yes: diet, alcohol consumption, exercise, and DM2.

rs328

rs322*

Yes

[19,29,33]

ABO

Regulates levels of lipids, adhesion molecules, vWF, and FVIII; affects lipid metabolism, inflammation, and thrombosis. In non-O groups, a prothrombotic state is favored, and it is associated with the presence and severity of PAD.

↑vWF and FVIII: ↑Thrombosis risk

Yes: smoking, hypertension, and coagulation factors.

rs505922*

rs616154*

rs635634

rs8176719

Yes

[29,33,38,83]

IL-6

The increase in IL-6 is associated with increased inflammation.

↑IL-6: ↑PAD and adverse cardiovascular events  risk

Yes: chronic inflammation, smoking, alcohol consumption, obesity, sedentary lifestyle, and DM2.

rs4722172*

rs1800795

rs2228145

rs2069827*

Contradictory results

[12,29,40,68,71,74,75,84–86]

SH2B3

Participates in the signaling of immune and hematopoietic cells, promotes inflammation, vascular tone, and PAD.

↑ PAD risk with risk alleles

Yes: hypertension and obesity.

rs3184504*

rs7528419

rs653178

Yes

[7,12,29,37,40,41,52,83,87]

CELSR2

Alters LDL metabolism, promoting lipid accumulation, inflammation and  atherosclerosis.

↑LDL: ↑Atherosclerosis risk

Yes: diet, dyslipidemia, and obesity.

rs12740374

No

[12,19,37,83]

HDAC9

Modulates inflammation and the VSMC phenotype and is associated with vascular calcification and atherosclerosis.

↑Expression/activity: ↑PAD risk

Yes: smoking, hypertension, and DM2.

rs2107595

rs2074633*

Yes

[7,29,37,48,66]

CDKN2B-AS1

Regulates the cell cycle, promotes inflammation and VSMC proliferation.

↑CDKN2B-AS1 expression: ↑ Atherosclerosis and PAD risk

Yes: smoking and DM2.

rs1537372*

rs10738610*

rs1333049

rs10757278

rs10757269

Yes

[29,33,37,48,50,52,88]

PTPN11

Participates in endothelial signaling, promotes inflammation, and is associated with pro-atherogenesis

↑Function: ↑PAD risk

Yes: hypertension and DM2.

rs11066301*

rs10774624

Yes

[12,19,29,33,37]

CREB3L1

Modulates the stress response, is associated with vascular calcification, and promotes cellular remodeling and inflammation.

Depends on the context

Yes: obesity and DM2.

rs7476*

Contradictory results

[7,19,29,37,47]

COL4A1

Participates in the integrity of the basement membrane; its genetic variants cause microvascular fragility.

Defect:

↑Fragility

Yes: smoking and hypertension.

rs1975514*

Contradictory results

[7,12,29,35,89]

SMOC1

Regulates cellular homeostasis, cell migration and proliferation, calcification, tissue fibrosis, and angiogenesis.

↑SMOC1: ↑Calcification/ cell remodeling

Yes: DM2.

rs55784307*

Contradictory results

[12,29,33,37,46]

MMP-3

Overexpression is associated with extracellular matrix degradation and vulnerability of the atherosclerotic plaque.

↑MMP3:

Instability/rupture of the atherosclerotic plaque

Yes: smoking, obesity, and DM2.

rs566125*

rs5030352

Yes

[7,29,47,68,75]

F5

The Factor V Leiden mutation causes resistance to protein C, which promotes a prothrombotic state in PAD.

↑Hypercoagulability

Yes: smoking, obesity, pregnancy, and contraceptives.

rs6025*

Contradictory results

[7,12,19,29,48,49,71,90]

F2

Mutations in prothrombin increase the risk of clots forming on the atherosclerotic plaque.

↑Activity: ↑Thrombotic risk

Yes: smoking, obesity, pregnancy, and contraceptives.

rs1799963

Contradictory results

[49,71,91–93]

FGB

Overexpression is associated with elevated plasma fibrinogen, inflammation, thrombosis and worse prognosis.

↑Expression: ↑Fibrinogen levels and this, ↑PAD and CVDs risk

Yes: smoking, alcohol consumption, inflammation, obesity, and DM2.

rs4220

rs1800790

Contradictory results

[5,48,94–97]

* Genetic variants have been replicated in multiple populations or meta-analyses. ABO: alpha 1-3-N-acetylgalactosaminyltransferase and alpha 1-3-galactosyltransferase; CDKN2B: Cyclin-dependent kinase inhibitor 2B-antisense 1; CELSR2: Cadherin EGF LAG seven-Pass G-type receptor 2; COL4A1: Collagen type IV alpha 1 chain; CREB3L1: CAMP responsive element binding protein 3 like 1; CVDs: Cardiovascular diseases; DM2: Type 2 diabetes mellitus; F2: Factor 2 (Prothrombin); F5: Factor 5; FGB: Fibrinogen beta chain; FVIII: Factor VIII; HDAC9: Histone deacetylase 9; HDL: High-density lipoprotein; ICAM1: Intercellular adhesion molecule 1; IL-6: Interleukin-6; LIPC: Lipase C, hepatic type; LDL: Low-density lipoprotein; LDLR: LDL receptor; Lp(a): Lipoprotein (a); LPL: Lipoprotein lipase; MMP3: Matrix metallopeptidase 3; NOS3: Nitric oxide synthase 3; PTPN11: Protein tyrosine phosphatase non-receptor type 11; sdLDL-C: Small dense low-density lipoprotein cholesterol; SELE: Selectin E; SH2B3: SH2B adaptor protein 3; VSMCs: Vascular smooth muscle cells; vWF: von Willebrand factor.

6a. I think that it would enrich the discussion section of the manuscript if the authors could add some aspects regarding new technology such as single-cell transcriptomics or CRISPR-based functional validation in order to enhance further our general understanding of PAD genetics.

 ANSWER:

We agree with the reviewer, thank you for your suggestion. The following information was added in the manuscript (LINES 580 - 591) and new references were included:

Furthermore, additional experimental strategies may be integrated in the research of gene functions by activating or silencing them. Therefore, gene editing in PAD is emerging as a promising but still developing and underexplored new field of research. Gene editing is being investigated through a gene therapy and genetic modification approach. This has made it an effective project for precisely modifying DNA. However, it is still an experimental therapy under development that could allow for the correction of genetic mutations or the silencing of harmful genes in the future, thus restoring vascular function in this group of patients. Although, this technology fits more for functional assay profiles; for example, recent studies have employed induced pluripotent stem cells as a model, utilizing CRISPR-Cas9 technology to introduce genetic variants into cell clones. This strategy might enable the functional validation of specific variants and the elucidation of their role as mechanisms in various complex diseases such as PAD [107–110].

7a. I felt the manuscript lacking figures and diagrams. Additional figures (such as a schematic overview of gene-pathway relationships in PAD, or a translational flowchart linking genetics to clinical application) would greatly enhance the accessibility, pedagogical value and robustness of the review. Considering the genetic-density of the data presented, smart visuals could help readers power through this Gordian molecular knot to better comprehend the links between the molecular mechanisms and the disease outcomes.

 ANSWER:

Thanks to the reviewer for this valuable contribution. Based on their recommendations, we have improved Figure 1 and added Figures 2 and 3. We have also incorporated Tables 2 and 3.

Below is a list of the changes implemented in the improved version of the review:

Figure 1. Gene classification by role in peripheral arterial disease.

Figure 2. PAD genes and their connection to KEGG pathways.

Figure 3. Overview of the integrative approach connecting polymorphisms, functional assays, and molecular pathways in PAD.

Table 1. Genetic variants identified in PAD.

Table 2. Genetic variants replicated in multiple studies associated with PAD and their relation with environmental and lifestyle factors, metabolic risks, and drugs.

Table 3. Factors that influence the failure to replicate findings in genetic association studies.

Suggested Additions:

Figure 2: A visual map of key genes and their roles across the stages of PAD (origination → progression → rupture).

ANSWER:

Thank you to the reviewer. This suggestion was taken into account in the revised version of the manuscript (LINES 439 - 444).

Figure 2. PAD genes and their connection to KEGG pathways.

 Figure 3: A translational roadmap showing how identified genes/SNPs could contribute to risk prediction, therapeutic targets, or biomarker development.

 ANSWER:

We agree with the reviewer. Now, we added in figure 3 with a translational roadmap as the reviewer suggestion. (LINES 570 –575).

Figure 3. Overview of the integrative approach connecting polymorphisms, functional assays, and molecular pathways in PAD.

 Comments on the Quality of English Language

Title and Language Quality

Title: “Genetic Basis into Peripheral Artery Disease” is grammatically incorrect. Suggestion:

→ "Genetic Basis of Peripheral Artery Disease" or

→ "Genetic Insights into Peripheral Artery Disease"

ANSWER:

Thank you to the reviewer for this important point. We agree with your suggestion. We changed the title according your observation.

The new title: “Genetics Insights into Peripheral Artery Disease: A Narrative Review”. (LINES 2 - 3)

Language & Grammar: The manuscript contains frequent grammatical issues, awkward constructions, and unnatural phrasing.

ANSWER:

We are in accordance with the reviewer. The manuscript was submitted for professional language editing to improve grammar, clarity, and overall flow. Additionally, we expanded specific sections to provide more detail, as recommended.

Reviewer 2 Report

Comments and Suggestions for Authors

This paper is devoted to the review of genetic factors of peripheral vascular disease (PAD). The authors analyzed and structured the known data on genes, the variants of which are associated with PAD. The paper lacks conclusions made with the help of the authors' valuable experience and based on the listed works; in addition to listing brief properties of genes and their variants, a more interesting analysis could have been made. In general, the work is done carefully, the material is presented easily and clearly. Taking into account the recommendations below, the authors can make the review more valuable and significant.

1) ‘In the evaluation of PAD, systematic lower-extremity examination is required, aiming to identify muscle or skin atrophy, loss of hair, and nail hypertrophy. Auscultation of bruits may be practical, but exhaustive assessment of peripheral pulses is indispensable [4,7]’ (lines 44-47)

‘This test is used to assess vascular health, especially to detect PAD. ABI A normal ABI value ranges from 1 to 1.4, borderline values are 0.91-0.99, and abnormal values are ≤ 0.9 for clinical and epidemiological purposes [4,6,7]’ (lines 49-52)

‘ABI is the standard method for diagnosing PAD; however, it is not reliable in patients with calcified noncompressible vessels, who tend to exhibit ABI values greater than 1.4. This is commonly seen in diabetics, and individuals with chronic kidney disease, who depend on the toe-brachial index (TBI) or more sophisticated imaging methods to establish an accurate diagnosis [4,7]’ (52-56)

- It is not clear why you referred to work [7] (David et al. (2021)); perhaps you mixed up the reference number.

2) ‘ABI’ (line 50) - There is an unnecessary abbreviation in the line.

3) ‘Thus, in this review article, we described an overview of the role of genetics implicated in the peripheral artery disease’ (lines 27-28 and lines 113-114) – It's better not to write the same text in Abstract and text body.

4) The reference list contains reviews on your topic (e.g. Ochoa Chaar et al. (2024) [32]); please explain what new information you have presented and what advantages your work has. In this context, it would be interesting to discuss in more detail the possibilities of treatment and prevention of PAD (e.g. for other cardiovascular diseases).

5) It would be better if you discussed the works dedicated to the study of gene expression in PAD.

6) It would be better if you discussed the gene expression studies in PAD. Also, could you give a more detailed overview of the molecular action of the identified genes and draw parallels with other vascular diseases?

Author Response

REVIEWER 2

We thank the reviewer for the insightful comments and helpful criticisms. We have responded to all comments as detailed below and we now hope that you will find our revised Manuscript acceptable for publication in “Biomedicines”.

Comments and Suggestions for Authors

 This paper is devoted to the review of genetic factors of peripheral vascular disease (PAD). The authors analyzed and structured the known data on genes, the variants of which are associated with PAD. In general, the work is done carefully, the material is presented easily and clearly. Taking into account the recommendations below, the authors can make the review more valuable and significant.

QUESTIONS:

1) In the evaluation of PAD, systematic lower-extremity examination is required, aiming to identify muscle or skin atrophy, loss of hair, and nail hypertrophy. Auscultation of bruits may be practical, but exhaustive assessment of peripheral pulses is indispensable [4,7]’ (lines 44-47).

- This test is used to assess vascular health, especially to detect PAD. ABI A normal ABI value ranges from 1 to 1.4, borderline values are 0.91-0.99, and abnormal values are ≤ 0.9 for clinical and epidemiological purposes [4,6,7]’ (lines 49-52)

- ABI is the standard method for diagnosing PAD; however, it is not reliable in patients with calcified noncompressible vessels, who tend to exhibit ABI values greater than 1.4. This is commonly seen in diabetics, and individuals with chronic kidney disease, who depend on the toe-brachial index (TBI) or more sophisticated imaging methods to establish an accurate diagnosis [4,7]’ (52-56)

- It is not clear why you referred to work [7] (David et al. (2021)); perhaps you mixed up the reference number.

ANSWER:

We agree with the reviewer, thank you for your kind observation. Now, in the revised version, we deleted the reference 7 in lines 48 – 52, and 52 – 56. As the reviewer rightly comments, we confused the reference number. Thanks again.

2) ‘ABI’ (line 50) - There is an unnecessary abbreviation in the line.

ANSWER:

Thanks to the reviewer. We deleted in the revised manuscript.

3) Thus, in this review article, we described an overview of the role of genetics implicated in the peripheral artery disease’ (lines 27-28 and lines 113-114) – It's better not to write the same text in Abstract and text body.

 ANSWER:

We are in accordance with the reviewer. Now, we have modified the text in the body of the manuscript, as it appears below:

TEXT BODY:

Thus, this narrative review presents a descriptive approach to identify the genetic de-terminants involved in the pathophysiology of PAD. We include the updated ESC 2024 PAD clinical guidelines to provide specific recommendations for appropriate disease management. We also identify the pathways involved and transcriptomics in genes that converge in PAD. Finally, we describe the relationship of genetic variants associated with PAD with environment, lifestyle, metabolic risk, and drugs (LINES 93 - 99).

 4a) The reference list contains reviews on your topic (e.g. Ochoa Chaar et al. (2024) [32]); please explain what new information you have presented and what advantages your work has.

 ANSWER:
We appreciate the reviewer's observation. While previous reviews such as Ochoa Chaar et al. (2024) have addressed general aspects of periphelaqueral artery disease (PAD), our manuscript: we describe the main genetic variants associated with p initiation, progression, and rupture in PAD. Furthermore, we identify different KEGG pathways involved in the pathological processes of these genes. We also describe gene expressions or transcriptomic studies, particularly in biopsies from patients with PAD. (LINES 26 – 30).

4b) In this context, it would be interesting to discuss in more detail the possibilities of treatment and prevention of PAD (e.g. for other cardiovascular diseases).

ANSWER:

Thanks to the reviewer. Now, we added this information in the perspectives section for this important point and we included the figure 3 to represents this issue (LINES 558 -575).

The integration of genetic insights into PAD management has important clinical implications, particularly for risk stratification and personalized prevention. A proposed clinical pathway begins with the identification of at-risk individuals, combining family history and conventional factors with polygenic risk scores when available [103]. In those at elevated genetic risk, early non-invasive vascular screening (ankle–brachial index, duplex ultrasound) may be warranted, followed by intensified preventive strategies, including optimized lipid, blood pressure, and glycemic control, as well as structured smoking cessation. Genetic insights may also guide patient selection for novel therapies or clinical trials targeting PAD-specific molecular mechanisms. This stepwise integration of genetic risk into vascular practice exemplifies the potential of precision medicine to improve outcomes in PAD [12]. (Figure 3).

Figure 3. Overview of the integrative approach connecting polymorphisms, functional assays, and molecular pathways in PAD, providing insights into disease mechanisms and guiding potential preventive and therapeutic approaches. (a) Medical assessment and identification of individual risk factors for PAD. (b) Integration of genetic variation analysis and functional assays to elucidate key genes and pathways involved in PAD pathogenesis. (c) A future application of findings to personalized strategies for PAD prevention. Created with Biorender.

5) It would be better if you discussed the works dedicated to the study of gene expression in PAD. 

ANSWER:

We agree with the reviewer suggestion. Now, we included this important information in the LINES 478 – 500.

In parallel, we describe gene expression or transcriptomic studies that are crucial for the study of PAD. These studies are relevant as they demonstrate how changes in transcriptional activity could contribute to disease and reveal information not available at the genomic level. It is important to note that studies of this nature at the tissue level are scarce in PAD.

In this sense, transcriptomic and proteomic analyses performed in humans on gastrocnemius muscle biopsies from patients with PAD and control participants without PAD showed several enriched pathways, including those related to hypoxia, such as phosphatase and tensin homolog (PTEN), phosphoinositide 3-kinase (PI3K), and mitogen-activated protein kinase (MAPK) signaling [63]

Furthermore, WNT, Hedgehog, and Notch are among the key signaling pathways involved in the repair of damage caused by chronic hypoxia or ischemia/reperfusion. In this regard, a prolonged hypoxic environment in muscle is associated with induced mitochondrial damage, reduced ATP production, and the stimulation of inflammatory responses, including NF-κB activation [63]. It has also been shown that samples from patients with PAD are characterized by increased inflammation and decreased glycolysis compared to control participants without PAD [64].

On the other hand, transcriptomic studies in human gastrocnemius muscle biopsies showed a global overexpression of genes involved in stress response, autophagy, hypoxia, and muscle atrophy, as well as deficiencies in angiogenic protein signaling, response to misfolded proteins, and nerve repair, which could contribute to poor limb function in PAD [64,65]. Each of these mechanisms has potential for future research to develop effective interventions for the treatment of PAD.

6) Could you give a more detailed overview of the molecular action of the identified genes and draw parallels with other vascular diseases?

ANSWER:

We appreciate the reviewer’s insightful suggestion. To address this point, we expanded the manuscript to include molecular mechanisms of the identified genes.

To further contextualize these findings, the PAD-associated genes involved in the origin, progression, and rupture of the plaque were scrutinized through ExpressAnalyst (web-based platform) for functional enrichment analysis to KEGG pathways, and the overlap of genes involved in each pathway is highlighted (LINES 428-477)

To highlight the relevance of the genes described in Table 1 associated with PAD, a KEGG pathway enrichment analysis was performed to represent the pathways involved in these genes' disease processes. In this regard, at the plaque formation stage, it allowed for the integration of the arginine biosynthesis pathway, the glycosphingolipid pathway, and the fluid shear stress and atherosclerosis pathway. Genes such as NOS3, ABO, and SELE are involved in these pathways (Figure 2A).

The importance of these pathways is as follows:

1) Arginine biosynthesis produces NO, a molecule that relaxes and dilates blood vessels, improving blood flow. In addition, L-arginine may improve walking ability and blood flow in these patients [54,55].

2) The biosynthesis of glycosphingolipids and their metabolites, when altered, can lead to their accumulation in tissues. This accumulation has been linked to the development and progression of PAD. It is suggested that alterations in sphingolipid metabolism contribute to the cellular and tissue damage that occurs during the atherosclerotic process [56].

3) Fluid shear stress and atherosclerosis, due to the force exerted by the constant flow of blood on the walls of blood vessels, play an important role in atherogenesis by altering the integrity of the endothelium, increasing its permeability, and allowing the entry of lipoproteins and inflammatory cells, which initiates the process of plaque formation and thrombi in the arteries and causes PAD [57].

Moreover, the pathways related to the stage of plaque progression include the PI3K-Akt signaling pathway, TNF signaling pathway, and insulin resistance pathway. Genes such as NOS3, IL-6, CREB3L1, PTPN11, and SELE are involved in these pathways (Figure 2B). The relevance of these pathways is as follows:

1) The PI3K-Akt signaling pathway is involved in regulating inflammation and oxidative stress, both of which are key factors in the pathogenesis of PAD [58].

2) The TNF signaling pathway promotes oxidative stress and decreases the bioavailability of NO, a crucial vasodilator, contributing to endothelial dysfunction. Furthermore, the TNF signaling pathway impacts vascular remodeling processes, leading to structural changes in the arteries of patients [59].

3) The insulin resistance pathway causes vascular damage through endothelial dysfunction via the inhibition of NO production. In addition, insulin resistance can activate pro-inflammatory molecular pathways, such as the MAP kinase (MAPK) pathway, which contribute to the disease [60].

Finally, the pathways related to the plaque rupture stage allow for the integration of complement activation, coagulation cascade, and platelet activation, which are involved in exacerbating the inflammatory process, which, together with endothelial dysfunction promotes thrombosis. Genes such as F5, F2, and FGB are involved in these pathways (Figure 2C) [61,62].

(a) Plaque origination

(b) Plaque progression

(c) Plaque rupture

Figure 2. PAD genes and their connection to KEGG pathways. The PAD-associated genes involved in the origin, progression, and rupture of plaque were scrutinized using ExpressAnalyst (a web-based platform) for functional enrichment analysis to KEGG pathways. The overlap of genes involved in each pathway is highlighted [53]. (a) KEGG pathways interaction of plaque origination. (b) KEGG pathways interaction of plaque progression. (c) KEGG pathways interaction of plaque rupture.

Reviewer 3 Report

Comments and Suggestions for Authors

The article addresses a relevant and timely topic by providing a useful synthesis of the genetics underlying peripheral artery disease.

The scope is clear.

The literature base appears adequate.

However, the manuscript would benefit from minor revisions. These would enhance clarity, grammar, and flow. They would also provide slightly more detail in certain areas. For example, including specific genetic mechanisms.

Author Response

REVIEWER 3

 We thank to the reviewer for the insightful comments and helpful criticisms. We have responded to all comments as detailed below and we now hope that you will find our revised Manuscript acceptable for publication in “Biomedicines”.

Comments and Suggestions for Authors

The article addresses a relevant and timely topic by providing a useful synthesis of the genetics underlying peripheral artery disease.

The scope is clear. The literature base appears adequate.

However, the manuscript would benefit from minor revisions. These would enhance clarity, grammar, and flow. They would also provide slightly more detail in certain areas. For example:

1) Including specific genetic mechanisms.

ANSWER:

We are in accordance with the reviewer. The manuscript was submitted for professional language editing to improve grammar, clarity, and overall flow. Additionally, we expanded specific sections to provide more detail, as recommended.

Furthermore, we elaborated on the contextualization of our findings by analyzing PAD-associated genes involved in plaque origin, progression, and rupture. These genes were examined using ExpressAnalyst (a web-based platform) for functional enrichment analysis through KEGG pathways, and the overlap of genes involved in each pathway was highlighted. This addition strengthens the manuscript’s coherence and enhances its contribution as a timely and focused synthesis of the genetic landscape of peripheral artery disease.

Next, we added this information in LINES 428- 477.

To highlight the relevance of the genes described in Table 1 associated with PAD, a KEGG pathway enrichment analysis was performed to represent the pathways involved in these genes' disease processes. In this regard, at the plaque formation stage, it allowed for the integration of the arginine biosynthesis pathway, the glycosphingolipid pathway, and the fluid shear stress and atherosclerosis pathway. Genes such as NOS3, ABO, and SELE are involved in these pathways (Figure 2A).

The importance of these pathways is as follows:

1) Arginine biosynthesis produces NO, a molecule that relaxes and dilates blood vessels, improving blood flow. In addition, L-arginine may improve walking ability and blood flow in these patients [54,55].

2) The biosynthesis of glycosphingolipids and their metabolites, when altered, can lead to their accumulation in tissues. This accumulation has been linked to the development and progression of PAD. It is suggested that alterations in sphingolipid metabolism contribute to the cellular and tissue damage that occurs during the atherosclerotic process [56].

3) Fluid shear stress and atherosclerosis, due to the force exerted by the constant flow of blood on the walls of blood vessels, play an important role in atherogenesis by altering the integrity of the endothelium, increasing its permeability, and allowing the entry of lipoproteins and inflammatory cells, which initiates the process of plaque formation and thrombi in the arteries and causes PAD [57].

Moreover, the pathways related to the stage of plaque progression include the PI3K-Akt signaling pathway, TNF signaling pathway, and insulin resistance pathway. Genes such as NOS3, IL-6, CREB3L1, PTPN11, and SELE are involved in these pathways (Figure 2B). The relevance of these pathways is as follows:

1) The PI3K-Akt signaling pathway is involved in regulating inflammation and oxidative stress, both of which are key factors in the pathogenesis of PAD [58].

2) The TNF signaling pathway promotes oxidative stress and decreases the bioavailability of NO, a crucial vasodilator, contributing to endothelial dysfunction. Furthermore, the TNF signaling pathway impacts vascular remodeling processes, leading to structural changes in the arteries of patients [59].

3) The insulin resistance pathway causes vascular damage through endothelial dysfunction via the inhibition of NO production. In addition, insulin resistance can activate pro-inflammatory molecular pathways, such as the MAP kinase (MAPK) pathway, which contribute to the disease [60].

Finally, the pathways related to the plaque rupture stage allow for the integration of complement activation, coagulation cascade, and platelet activation, which are involved in exacerbating the inflammatory process, which, together with endothelial dysfunction promotes thrombosis. Genes such as F5, F2, and FGB are involved in these pathways (Figure 2C) [61,62].

(a) Plaque origination

(b) Plaque progression

(c) Plaque rupture

Figure 2. PAD genes and their connection to KEGG pathways. The PAD-associated genes involved in the origin, progression, and rupture of plaque were scrutinized using ExpressAnalyst (a web-based platform) for functional enrichment analysis to KEGG pathways. The overlap of genes involved in each pathway is highlighted [53]. (a) KEGG pathways interaction of plaque origination. (b) KEGG pathways interaction of plaque progression. (c) KEGG pathways interaction of plaque rupture.

Reviewer 4 Report

Comments and Suggestions for Authors

Dear editor,

I reviewed the article entitled “Genetic Basis into Peripheral Artery Disease”. the manuscript focus on an important issue. It is well written and easy to follow. I congrats the authors for their effort. I think, the paper provides significant contribution to the literature. I have only minor comments:

-ESC 2024 peripheral artery disease guideline provide some specific recommendations. Please include them into your review with their class/level of evidence.

Sincerely.

Author Response

REVIEWER 4

We thank to the reviewer for the insightful comments and helpful criticisms. We have responded to all comments as detailed below and we now hope that you will find our revised Manuscript acceptable for publication in “Biomedicines”.

Comments and Suggestions for Authors

I reviewed the article entitled “Genetic Basis into Peripheral Artery Disease”. the manuscript focus on an important issue. It is well written and easy to follow. I congrats the authors for their effort. I think, the paper provides significant contribution to the literature. I have only minor comments:

1) ESC 2024 peripheral artery disease guideline provide some specific recommendations. Please include them into your review with their class/level of evidence.

ANSWER:

We appreciate the reviewer's valuable comments. In response, we have included a paragraph summarizing the key recommendations from the 2024 ESC Guidelines on the treatment of peripheral arterial and aortic diseases.

The following paragraph was added in section 2. Management of PAD in ESC 2024 Guidelines (LINES 101-117).

The 2024 ESC Guidelines on the Management of Peripheral Arterial and Aortic Diseases (PAAD) underscore PAD as a chronic, progressive atherosclerotic condition associated with a significantly increased risk of cardiovascular events and limb-related complications. The guidelines emphasize the importance of early diagnosis, particularly through ABI screening in high-risk populations [Class I, Level B] as well as the comprehensive management of modifiable risk factors, including dyslipidemia, hypertension, DM2, and smoking [Class I, Level A]. Structured exercise therapy is also recommended as a core component of conservative treatment [Class I, Level A evidence]. Long-term follow-up within a multidisciplinary care framework is advised [Class I, Level C], and revascularization either endovascular or surgical should be considered in patients with critical limb-threatening ischemia (CLTI) [Class I, Level B] or non-healing ulcers. The need for revascularization and the risk of major amputation may now be stratified using validated clinical tools such as the WIfI (Wound, Ischemia, and foot Infection) classification system [Class IIa, Level C]. Future predictive models may now be stratified using vali-dated clinical tools such as the WIfI classification system [Class IIa, Level C]. Future predictive models may incorporate genetic profiling to further refine risk stratification and personalize therapeutic decision making in PAD management [2].

Reviewer 5 Report

Comments and Suggestions for Authors

Dear Editor,

I evaluated the paper and I think that:

  • Authors should revise the Introduction section:
  • a. the lenght of this section should be reduced.
  • b. please be more focused on the main aims of the paper
  • c. the final paragraph ("in summary....Thus...") should be revised as there is no need for resuming previous data already discussed in the Introduction section, but rather the authors should amplify the description of the aims of the paper
  • according to me, there is a specific bias in this analysis. The authors tried to evaluate the genetic background of PAD. Indeed, genetics is related to atherosclerosis. They should outline how to differentiate the "specific" genetic pattern for PAD rather than systemic atherosclerosis.
  • Authors should indicate the clinical implications of their analysis. A clinical decision pathway should be described and included.
  • similarly, please define the role of gene editing in this setting

Author Response

REVIEWER 5

We thank to the reviewer for the insightful comments and helpful criticisms. We have responded to all comments as detailed below and we now hope that you will find our revised Manuscript acceptable for publication in “Biomedicines”.

We are in accordance with the reviewer. The manuscript was submitted for professional language editing to improve grammar, clarity, and overall flow.

Comments and Suggestions for Authors

I evaluated the paper and I think that:

Authors should revise the Introduction section:

  1. The length of this section should be reduced.

ANSWER:

We agree with the reviewer. We acknowledge the suggestion and we have taking into account the point.

  1. b) Please be more focused on the main aims of the paper 

ANSWER:

Thank you to the reviewer for this important suggestion. Now, we focus on the main objectives and they were described in the revised version in the introduction section (LINES 93 - 99):

Thus, this narrative review presents a descriptive approach to identify the genetic determinants involved in the pathophysiology of PAD. We include the updated ESC 2024 PAD clinical guidelines to provide specific recommendations for appropriate disease management. We also identify the pathways involved and transcriptomics in genes that converge in PAD. Finally, we describe the relationship of genetic variants associated with PAD with environment, lifestyle, metabolic risk, and drugs.

 c) The final paragraph ("in summary....Thus...") should be revised as there is no need for resuming previous data already discussed in the Introduction section, but rather the authors should amplify the description of the aims of the paper according to me, there is a specific bias in this analysis.

ANSWER:

We fully agree with the reviewer on this issue. As pointed in the previews question we clarify the final paragraph, and expanded the objectives of the review (LINES 93-99).

  1. d) The authors tried to evaluate the genetic background of PAD. Indeed, genetics is related to atherosclerosis. They should outline how to differentiate the "specific" genetic pattern for PAD rather than systemic atherosclerosis.

ANSWER:

We acknowledge to the reviewer for this important distinction. To identify PAD-specific patterns beyond general atherosclerosis, several strategies have been suggested:

In order to clarify this important point, we added the following paragraph (LINES 118 - 132).

On the other hand, it is important to differentiate the genetic pattern for PAD rather than atherosclerosis. Recent studies emphasize the importance of differentiating PAD-specific genetic signatures from those driving coronary or cerebrovascular disease. Several methodological approaches may help delineate this specificity, including case-only analyses in PAD patients without CAD or stroke, conditional GWAS adjusting for shared atherosclerotic risk, and tissue-specific expression studies in peripheral arteries [19]. Transcriptomic and expression quantitative trait locus analyses suggest that certain variants influence gene expression in limb vasculature more strongly than in coronary or cerebral beds, pointing to localized mechanisms such as impaired angiogenesis and collateral formation [12]. Importantly, PAD progression is shaped by unique biomechanical and hemodynamic factors in peripheral circulation, such as shear stress, collateral vessel development, and ischemia–reperfusion dynamics, which may account for functional differences in the genetic determinants of disease expression. Thus, PAD genetics are characterized by both shared systemic pathways and modifiers with context-specific effects in the peripheral vasculature.

  1. e) Authors should indicate the clinical implications of their analysis. A clinical decision pathway should be described and included.

ANSWER:

Thanks to the reviewer. Now, we added this information in the perspectives section for this important point and we included the figure 3 to represents this issue (LINES 558 -575).

The integration of genetic insights into PAD management has important clinical implications, particularly for risk stratification and personalized prevention. A proposed clinical pathway begins with the identification of at-risk individuals, combining family history and conventional factors with polygenic risk scores when available [103]. In those at elevated genetic risk, early non-invasive vascular screening (ankle–brachial index, duplex ultrasound) may be warranted, followed by intensified preventive strategies, including optimized lipid, blood pressure, and glycemic control, as well as structured smoking cessation. Genetic insights may also guide patient selection for novel therapies or clinical trials targeting PAD-specific molecular mechanisms. This stepwise integration of genetic risk into vascular practice exemplifies the potential of precision medicine to improve outcomes in PAD [12]. (Figure 3).

Figure 3. Overview of the integrative approach connecting polymorphisms, functional assays, and molecular pathways in PAD, providing insights into disease mechanisms and guiding potential preventive and therapeutic approaches. (a) Medical assessment and identification of individual risk factors for PAD. (b) Integration of genetic variation analysis and functional assays to elucidate key genes and pathways involved in PAD pathogenesis. (c) A future application of findings to personalized strategies for PAD prevention. Created with Biorender.

  1. f) Similarly, please define the role of gene editing in this setting

ANSWER:

We agree with the reviewer, thank you for your suggestion. The following information was added in the manuscript (LINES 580 - 591) and new references were included:

Furthermore, additional experimental strategies may be integrated in the research of gene functions by activating or silencing them. Therefore, gene editing in PAD is emerging as a promising but still developing and underexplored new field of research. Gene editing is being investigated through a gene therapy and genetic modification approach. This has made it an effective project for precisely modifying DNA. However, it is still an experimental therapy under development that could allow for the correction of genetic mutations or the silencing of harmful genes in the future, thus restoring vascular function in this group of patients. Although, this technology fits more for functional assay profiles; for example, recent studies have employed induced pluripotent stem cells as a model, utilizing CRISPR-Cas9 technology to introduce genetic variants into cell clones. This strategy might enable the functional validation of specific variants and the elucidation of their role as mechanisms in various complex diseases such as PAD [107–110].

Round 2

Reviewer 1 Report

Comments and Suggestions for Authors

Congratulation to the authors, I suppose it was a little hardwork to answer all the questions and adapt the manuscript according to most of the suggestions, but I do believe that the manuscript is mostly ready to be published.

Comments on the Quality of English Language

The language is clear except for some parts that concordance makes understanding a bit confused. I suggest a quick review of the grammar for an even clearer English

Reviewer 2 Report

Comments and Suggestions for Authors

The authors have done a great deal of careful work, not limited by the reviewer's comments. Their thoughtfulness and genuine desire to improve the manuscript's content are palpable: the title was changed, the abstract was emphasized correctly, and a significant chapter was added. The Conclusions chapter, which could have been shorter, was also rewritten. I would especially like to highlight the added tables and figures, which greatly clarify the review and could be very useful for other researchers. Thus, the article is ready for publication. Below are the reviewer's responses, point by point.

1-3) The comments have been taken in account.

4a) I thank the authors for their response and I also note that the abstract was thoroughly revised taking this comment into account.

4b) Thanks be to the authors for the wonderful revision of the Perspectives chapter, especially the illustration. Your vision for the future is inspiring.

5-6) Your additions are valuable and interesting.

Reviewer 5 Report

Comments and Suggestions for Authors

authors well addressed previous comments